# Turning copper into an efficient and stable CO evolution catalyst beyond noble metals

Jing Xue[1,2,6], Xue Dong[3,6], Chunxiao Liu [1], Jiawei Li [1], Yizhou Dai[1], Weiqing Xue[1], Laihao Luo[1], Yuan Ji[1], Xiao Zhang [4], Xu Li[1], Qiu Jiang [1], Tingting Zheng [1], Jianping Xiao [3,5] ✉ & Chuan Xia [1] ✉

Using renewable electricity to convert $CO_2$ into CO offers a sustainable route to produce a versatile intermediate to synthesize various chemicals and fuels. For economic $CO_2$-to-CO conversion at scale, however, there exists a trade-off between selectivity and activity, necessitating the delicate design of efficient catalysts to hit the sweet spot. We demonstrate here that copper co-alloyed with isolated antimony and palladium atoms can efficiently activate and convert $CO_2$ molecules into CO. This trimetallic single-atom alloy catalyst ($Cu_{92}Sb_5Pd_3$) achieves an outstanding CO selectivity of 100% (±1.5%) at −402 mA cm$^{-2}$ and a high activity up to −1 A cm$^{-2}$ in a neutral electrolyte, surpassing numerous state-of-the-art noble metal catalysts. Moreover, it exhibits long-term stability over 528 h at −100 mA cm$^{-2}$ with an $FE_{CO}$ above 95%. *Operando* spectroscopy and theoretical simulation provide explicit evidence for the charge redistribution between Sb/Pd additions and Cu base, demonstrating that Sb and Pd single atoms synergistically shift the electronic structure of Cu for CO production and suppress hydrogen evolution. Additionally, the collaborative interactions enhance the overall stability of the catalyst. These results showcase that Sb/Pd-doped Cu can steadily carry out efficient $CO_2$ electrolysis under mild conditions, challenging the monopoly of noble metals in large-scale $CO_2$-to-CO conversion.

The ever-growing energy demand and reliance on fossil fuels have resulted in a vicious cycle of increasing $CO_2$ emissions, which poses a grave threat to the global environment and climate. To break this cycle and achieve a circular economy, electrochemical reduction of $CO_2$ ($CO_2$RR) offers a promising solution that can utilize renewable electricity to convert $CO_2$ into valuable chemicals and fuels. Among the diversified products of $CO_2$RR, carbon monoxide (CO) stands out as a particularly attractive product because it offers a very high economic return per mole of electrons consumed[1] and can serve as a versatile building block for synthesizing various organic compounds and liquid fuels *via* Fischer-Tropsch synthesis. However, pursuing high CO selectivity and activity in the $CO_2$RR is challenging due to the intricate reaction pathways and fierce competition from the hydrogen evolution reaction (HER). Noble metal catalysts, such as gold (Au) and silver (Ag), exhibit outstanding CO production performance with low onset potentials and high CO selectivity. However, their high cost and scarcity limit their industrial viability. Moreover, these noble metals tend to be inactive and susceptible to HER under high production rates (Supplementary Fig. 1), undermining the CO selectivity. To achieve economical and scalable $CO_2$-to-CO conversion, ongoing effort is

[1]School of Materials and Energy, University of Electronic Science and Technology of China, Chengdu 611731, P. R. China. [2]Hefei National Research Center for Physical Sciences at the Microscale, University of Science and Technology of China, Hefei, Anhui 230026, P. R. China. [3]State Key Laboratory of Catalysis, Dalian Institute of Chemical Physics, Chinese Academy of Sciences, Dalian 116023, P. R. China. [4]Department of Mechanical Engineering, Research Institute for Advanced Manufacturing, The Hong Kong Polytechnic University, Hung Hom, Kowloon, Hong Kong SAR 999077, P. R. China. [5]University of Chinese Academy of Sciences, Beijing 100049, P.R. China. [6]These authors contributed equally: Jing Xue, Xue Dong. ✉e-mail: xiao@dicp.ac.cn; chuan.xia@uestc.edu.cn

underway to find cost-effective catalysts that can delicately harmonize the key parameters, e.g., low overpotential, high current density, high selectivity, and long durability.

Copper (Cu) is a unique metal that can significantly activate $CO_2$ and produce a variety of products, including hydrocarbons and oxygenates[2]. However, pristine Cu suffers from poor selectivity, especially for mono-carbon products such as CO and formate. To address this issue, Cu-based single-atom alloys (SAAs) have been developed in recent years to improve the selectivity for mono-carbon products in the $CO_2RR$. By alloying with single-atom metals, the electronic structure of the Cu metal base can be fine-tuned, which leads to an optimal balance of desorption and adsorption rates of both the reactants and intermediates, resulting in a desired selectivity of a specific product. For example, our previous work showed that lowering the *d*-band state of Cu active sites near isolated dopants weakens the binding strength of CO* and enhances $CO_2$ adsorption/activation[3–5]. This substantially reduces the probability of CO*-CO* coupling and inhibits the formation of $C_{2+}$ products. Otherwise, Cu catalysts have been demonstrated to be highly mobile under the $CO_2RR$ environment[6–8], which will form active nanograins by in situ structural evolution. While single-atom alloying could stabilize the surface neighboring Cu atoms to some degree, nevertheless, these binary SAAs also undergo alloy reconstruction under high production rates (typically > −500 mA cm$^{-2}$) due to a strong driving bias and their extremely low content of isolated dopants, creating an activity-stability dilemma for CO production[4,9].

In view of these obstacles, we posit that to promote CO selectivity and activity while achieving long-term stability, introducing more than one kind of single-atom metal into Cu would be a solution to the CO production dilemma, with more tuning knobs and dimensions to adjust the properties of SAAs. Of note, according to the thermodynamic relationship of free energy ($\Delta G = \Delta H - T\Delta S$), the mixing entropy ($\Delta S_{mix}$) of the system can increase as the number of elements in the alloy increases (see Supplementary Note), thereby leading to a lower $\Delta G$ and improved stability. In this work, we therefore describe a trimetallic alloy catalyst ($Cu_{92}Sb_5Pd_3$) that combines a copper metal base with two single-atom metal additions, antimony (Sb) and palladium (Pd). These single atoms act synergistically to shift the electronic structure of Cu to favor CO production and stifle the HER but also improve the stability of the catalyst by preventing atom aggregation. As a result, these trimetallic alloys delivered outstanding CO current densities of *ca.* −400 and −840 mA cm$^{-2}$ under low applied potentials of −0.93 (±0.03) and −1.27 (±0.04) V *vs.* a reversible hydrogen electrode (RHE), respectively, with excellent CO selectivity. It also maintained a long-term stability up to 22 days (528 h) at −100 mA cm$^{-2}$ with an $FE_{CO}$ higher than 95%. These performances surpass most of the state-of-the-art noble metal catalysts reported thus far.

## Results and discussion

We synthesized the trimetallic catalyst (Cu-Sb-Pb) using a co-reduction method in pure ethanol solution instead of deionized water (see Methods). This eliminated the need for exotic complexants such as citric acid since $Sb^{3+}$ would not precipitate in nonaqueous solvents such as ethanol[10,11], thus avoiding potential contaminants. Inductively coupled plasma atomic emission spectroscopy (ICP-AES) measurements revealed that the Sb and Pd contents in the as-prepared sample were *ca.* 5.0 and 3.0 at%, respectively. X-ray photoelectron spectroscopy (XPS) also demonstrated the successful incorporation of two metal components, Pd and Sb, into this trimetallic catalyst (Supplementary Fig. 2). The X-ray diffraction pattern of the as-synthesized catalyst showed a pure Cu crystal structure (PDF 04-0836, Supplementary Fig. 3), ruling out the formation of either Sb or Pd nanoparticles and verifying that the bulk phase alloy remained unoxidized. The morphology of the sample was characterized by transmission electron microscopy (TEM) with sizes ranging from 10 to 20 nm

(Supplementary Fig. 4). The atomic structure of the Cu-Sb-Pb catalyst was then investigated by high-angle annular dark-field scanning transmission electron microscopy (HAADF-STEM) combined with energy-dispersive X-ray spectroscopy (EDS). Figure 1a clearly reveals the atomic dispersion of Pd/Sb atoms across the Cu matrix, which are marked by yellow circles and magnified into a three-dimensional structure. Then, STEM-EDS mapping further confirmed an even distribution of Sb and Pd in the Cu base without noticeable aggregation. Additionally, large-scale EDS mapping also precluded the existence of Sb or Pd particles (Supplementary Fig. 5). The above results, taken together, demonstrate the successful synthesis of trimetallic alloys, namely, the $Cu_{92}Sb_5Pd_3$ catalyst.

To better comprehend how Sb and Pd atoms are arranged in the Cu base, we performed extended X-ray absorption fine structure (EXAFS) measurements to examine their coordination environment. Figure 1b, c show the EXAFS curves of Sb and Pd, respectively, for $Cu_{92}Sb_5Pd_3$. The peak at ~2.30 Å was attributed to the Sb-Cu bond, while no Sb-Sb bonds were detected, confirming the singly dispersed Sb atoms in the alloy (Fig. 1b). The wavelet transform (WT) of Sb *K*-edge EXAFS supports this finding, which displays only one intensity maximum at ~8.6 Å$^{-1}$ corresponding to Sb-Cu coordination (Fig. 1d and Supplementary Table 1). Similarly, the curve for Pd exhibits a peak at ~2.27 Å, which corresponds to the Pd-Cu bond, and no Pd-Pd were observed (Fig. 1c). This suggests that Pd atoms are also dispersed as single sites in the alloy. The WT of Pd *K*-edge EXAFS corroborates this result by showing an intensity maximum at ~9.6 Å$^{-1}$ corresponding to Pd-Cu coordination (Fig. 1d and Supplementary Table 2). Neither Sb-O nor Pd-O bonds were detected in the EXAFS profiles, implying that $Cu_{92}Sb_5Pd_3$ is not oxidized. This is also confirmed by the Cu *K*-edge EXAFS and WT results (Supplementary Figs. 6 and 7), which show only Cu-Cu bonds (~2.24 Å in the EXAFS profile) and no evidence of copper oxides. Of note, we did not observe the formation of the Sb-Pd motif, implying that the Sb and Pd dopants are highly diluted as isolated atoms by the Cu base. Based on these results, we can conclude that we successfully synthesized a trimetallic single-atom alloy, $Cu_{92}Sb_5Pd_3$.

To investigate the electronic interaction between the Cu base and the dopants under reaction conditions, we conducted an *operando* X-ray absorption spectroscopy (XAS) study. Note that due to the low contents and therefore weak signal intensities, we, unfortunately, failed to detect the Sb and Pd signals in situ. However, *operando* Cu *K*-edge X-ray absorption fine structure (XAFS) showed that the Cu matrix of $Cu_{92}Sb_5Pd_3$ maintained a higher oxidation state of Cu at the open circuit potential (OCP), probably caused by oxidation during the electrode preparation process, which could be immediately reduced to a nearly metallic state under cathodic potentials. Very interesting, the *operando* XAS analysis provides unambiguous experimental evidence that the Cu matrix's electronic state of $Cu_{92}Sb_5Pd_3$ presented partially electron-deficient states during the whole reaction (Fig. 1e and Supplementary Fig. 8), which could be ascribed to the charge redistribution between Sb/Pd additions and the Cu matrix. This observation implies that such a $Cu_{92}Sb_5Pd_3$ single-atom alloy with a different electronic structure will mediate the $CO_2$ conversion in a unique way compared to pure Cu.

To evaluate the $CO_2RR$ catalytic performance of $Cu_{92}Sb_5Pd_3$, we performed $CO_2$ electrolysis in a standard three-electrode flow cell system with 0.5 M $KHCO_3$ as the electrolyte (see Methods). Gas products were analysed using gas chromatography (GC), whereas ion chromatography (IC) and nuclear magnetic resonance (NMR) spectroscopy were employed for liquid product quantification. The NMR results showed that formate was the only solution-phase product (Fig. 2a and Supplementary Fig. 9), while the GC analysis detected CO and $H_2$ as major gas-phase products (Supplementary Fig. 10). As shown in Fig. 2a, b, a high plateau of $FE_{CO}$ over 95% was retained across a broad potential range from −0.78 (±0.02) to −1.09 (±0.03) V *vs.* RHE, whereas the competitive HER was suppressed to below 3%. The

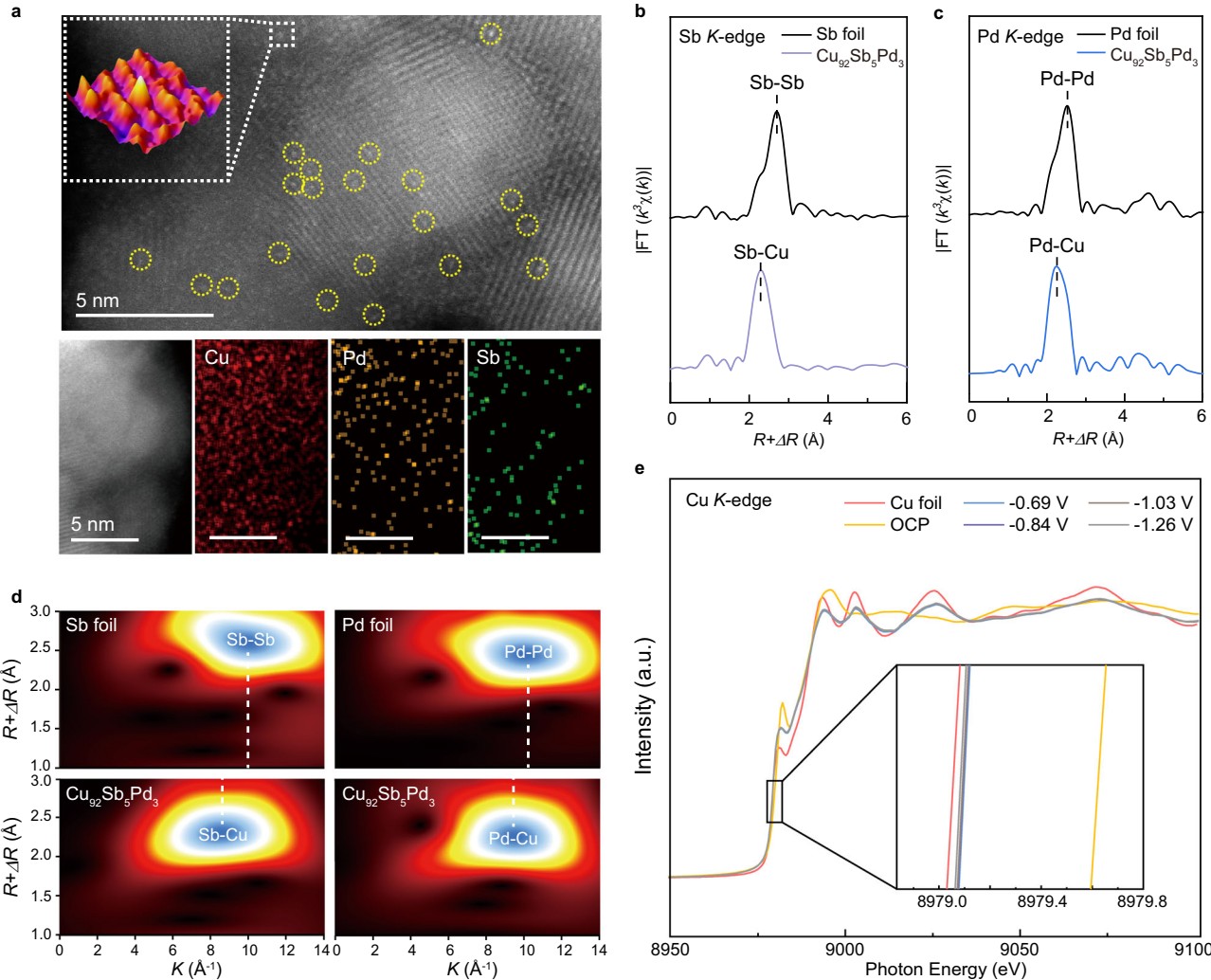

**Fig. 1 | Structural characterization of the Cu₉₂Sb₅Pd₃ catalyst. a** HAADF-STEM image and STEM-EDS mapping of Cu, Sb and Pd of the Cu₉₂Sb₅Pd₃ catalyst. The yellow circles highlight single Sb/Pd atoms, one of which was magnified into a 3D structure. Note that since the atomic numbers of Sb and Pd are quite close, HAADF-STEM failed to differentiate them. **b, c** Ex situ EXAFS spectra at the Sb and Pd *K*-edge of the Cu₉₂Sb₅Pd₃ catalyst, respectively. The spectra of Sb and Pd foil are shown as references. **d** EXAFS wavelet transforms for the Sb and Pd *K*-edge of the Cu₉₂Sb₅Pd₃ catalyst. Sb and Pd foil are shown as references. **e** *Operando* Cu *K*-edge XAFS spectra of the Cu₉₂Sb₅Pd₃ catalyst under applied potentials during the CO₂RR. Cu foil is shown as a reference. All potentials were calibrated to the RHE scale.

maximal $FE_{CO}$ reached up to 100% (±1.5%) with a CO partial current density ($j_{CO}$) of −402 mA cm⁻² at approximately −0.93 (±0.03) V *vs.* RHE. Notably, at approximately −1.19 (±0.04) V *vs.* RHE, Cu₉₂Sb₅Pd₃ delivered a high $j_{CO}$ exceeding −700 mA cm⁻² while still maintaining a CO selectivity of 90% (±2.8%). Moreover, 85% (±3.8%) $FE_{CO}$ could be sustained when the current density increased to −1000 mA cm⁻². To demonstrate that the exclusive selectivity for CO was due to the synergistic effect of both Pd and Sb single-atom components in Cu, binary single-atom alloy systems, namely, Cu₉₅Sb₅ and Cu₉₇Pd₃, and pure Cu nanoparticles were included for comparison using a similar method for Cu₉₂Sb₅Pd₃ (Supplementary Figs. 11–15 and Supplementary Table 3). In contrast to Cu₉₂Sb₅Pd₃, the CO₂RR catalytic performances of Cu₉₅Sb₅, Cu₉₇Pd₃, and pristine Cu were much less satisfying. Compared with pristine Cu, alloying either Sb or Pd single atoms could promote the selectivity and activity of CO to some extent (Fig. 2a, b and Supplementary Fig. 16). However, some C₂₊ products, such as C₂H₄ and alcohols, were also noticeable, especially under high production rates. In addition, the HER became dominant under high overpotentials, leading to a retarded increase in $j_{CO}$. To account for the influence of different electrochemically active surface area (ECSA) of the four catalysts, we normalized $j_{CO}$ by ECSA to compare their

intrinsic activities (Fig. 2c and Supplementary Fig. 17). The results showed that surface normalization exerts only a negligible effect on the performance trend. Additionally, we also increased the contents of Sb and Pd in the bimetallic counterparts, namely, Cu₉₂Sb₈ and Cu₉₂Pd₈ (Supplementary Table 3), to verify whether simply enhancing one single-atom composition could achieve such performance. The results in Supplementary Fig. 18 show that Cu₉₂Sb₈ produced a large amount of formate even under modest current densities, while Cu₉₂Pd₈ failed to suppress C-C coupling on the Cu matrix. Hence, we concluded that merely adding one single-atom component to the Cu base was insufficient, especially under a high current density, to achieve a high selectivity toward CO. The stunning catalytic performance of the trimetallic single-atom alloy Cu₉₂Sb₅Pd₃ stemmed from the concurrent presence of both Pd and Sb single-atom additions.

To decipher the effect of alloying both Sb and Pd single atoms on the HER, a major side reaction of the CO₂RR, we conducted cyclic voltammetry (CV) investigations to monitor hydrogen desorption peaks in the double layer region. Figure 2d shows that Cu₉₂Sb₅Pd₃ exhibited no hydrogen desorption peaks, unlike pristine Cu, which had prominent peaks indicating abundant hydrogen from the HER. This confirmed the suppression of the HER by introducing Pd and Sb atoms.

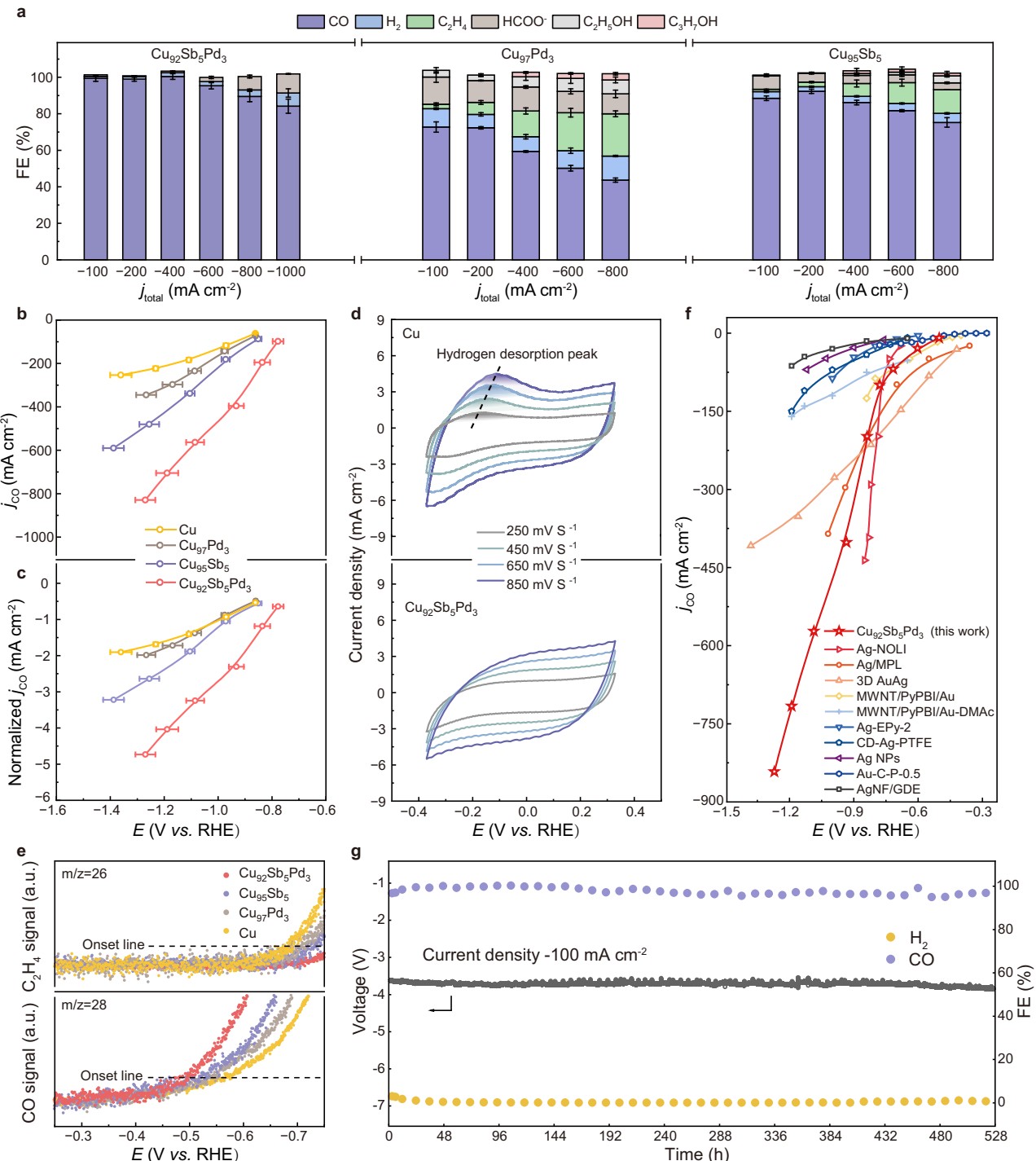

**Fig. 2 | CO₂RR performance over Cu₉₂Sb₅Pd₃ and control samples (Cu, Cu₉₅Sb₅ and Cu₉₇Pd₃).** **a** FEs of all CO₂RR products at different current densities for Cu₉₂Sb₅Pd₃, Cu₉₅Sb₅ and Cu₉₇Pd₃. **b, c** $j_{CO}$-V and ECSA normalized $j_{CO}$-V curves of four as-synthesized catalysts. The error bars in **a**–**c** correspond to the standard deviation of three independent measurements with 0.5 M KHCO₃ as the electrolyte. **d** CV investigations of the hydrogen desorption of the Cu and Cu₉₂Sb₅Pd₃ catalysts. **e** In situ DEMS measurements of four different catalysts in the CO₂RR. **f** $j_{CO}$-V curves of state-of-the-art noble metal catalysts in flow cell systems during the CO₂RR

compared with the Cu₉₂Sb₅Pd₃ catalyst. Catalyst references reproduced from Ag-NOLI (1 M KHCO₃)[50], Ag/MPL (0.1 M KHCO₃)[51], 3D AuAg (1 M KHCO₃)[52], MWNT/PyPBI/Au (2 M KHCO₃)[53], MWNT/PyPBI/Au-DMAc (1 M KCl)[54], Ag-EPy-2 (0.1 M KHCO₃)[55], CD-Ag-PTFE (1 M KHCO₃)[56], Ag NPs (2 M KHCO₃)[57], Au-C-P-0.5 (1 M KHCO₃)[58] and AgNF/GDE (1 M KCl)[59]. All potentials were calibrated to the RHE scale. **g** Stability test at −100 mA cm⁻² current density in MEA for 22 days (528 h) without iR corrections to the voltage.

We further investigated the possible reaction mechanism for CO₂-to-CO conversion on four different electrocatalysts using kinetic analysis. Tafel analysis was conducted to examine the rate determining steps (RDSs) involved in CO₂RR. The Tafel result plotted in Supplementary Fig. 19 revealed a faster kinetic process of CO formation on Cu₉₂Sb₅Pd₃

(138.7 mV dec⁻¹) than on the other three counterparts (Cu as 237.6 mV dec⁻¹, Cu₉₇Pd₃ as 211.2 mV dec⁻¹ and Cu₉₅Sb₅ as 199.8 mV dec⁻¹), indicating an accelerated electron transfer process[12,13]. When increasing the overpotential, a faster increase in the CO₂ reduction rate occurred on Cu₉₂Sb₅Pd₃, highlighting the critical role of

two single-atom metal components in boosting $CO_2$-to-CO conversion. Moreover, the Tafel slope of 138.7 mV dec$^{-1}$ for $Cu_{92}Sb_5Pd_3$ suggested that the first electron transfer step of $^*CO_2$ was the RDS[14]. Note that the deviation from a theoretical value of 118 mV dec$^{-1}$ (Supplementary Table 4) was likely due to more complicated electron transfer and electrochemical processes in real reactions[15]. Furthermore, the comparison of in situ differential electrochemical mass spectrometry (DEMS) results verified the promoted $CO_2$ reduction rate on $Cu_{92}Sb_5Pd_3$. Figure 2e shows that $Cu_{92}Sb_5Pd_3$ had a lower onset potential for CO generation but a higher onset potential for $C_2H_4$ formation than the other three, underlying its merit of inhibiting CO$^*$-CO$^*$ coupling to $C_{2+}$ products[16]. The obvious differences in onset potentials also revealed a successful modulation of Cu *via* alloying with two other single-atom metal components.

To elucidate the role of the two single-atom components in enhancing the $CO_2$RR, we benchmarked our catalyst against state-of-the-art noble metal catalysts in neutral electrolytes, *e.g.*, $KHCO_3$ or KCl. As illustrated in Fig. 2f and Supplementary Fig. 20, these noble metal catalysts exhibit similar CO onset potentials, but their current densities are far from meeting the requirements for industrial applications. At more negative potentials, their faradic efficiencies and partial current densities for CO plummet rapidly due to overwhelming HER[17,18]. In comparison, $Cu_{92}Sb_5Pd_3$ is on par with or even surpasses noble metals in terms of selectivity but also attained an extremely high current density that outshines most noble metal catalysts. To evaluate the durability of $Cu_{92}Sb_5Pd_3$ under realistic conditions, a long-term stability test was conducted in a membrane electrode assembly (MEA) at a current density of $-100$ mA cm$^{-2}$. Strikingly, the results show that the $FE_{CO}$ was maintained above 95% for 22 days without an evident voltage drop (Fig. 2g). In particular, the robust durability of $Cu_{92}Sb_5Pd_3$ even outperforms previously reported state-of-the-art noble metal catalysts (Supplementary Table 5). The outstanding durability was supposed to be derived from the increased mixed entropy of the trimetallic system that improved the stability by suppressing atom aggregation. Post-catalysis analyses, combining HAADF-STEM, STEM-EDS, and large-scale EDS screening (Supplementary Figs. 21-23), all demonstrated well-dispersed Sb/Pd atoms on the Cu matrix after $CO_2$RR, further attesting the robust durability of $Cu_{92}Sb_5Pd_3$. In comparison, the bimetallic counterpart, $Cu_{95}Sb_5$, which showcased a considerable improvement in CO generation, failed to maintain the SAA structure after a large current density electrolysis. The HAADF-STEM and EDS figures in Supplementary Fig. 24 exhibit the segregation of the Sb composition after the $CO_2$RR over $-800$ mA cm$^{-2}$, demonstrating the inferior stability of the bimetallic counterparts. Such a phenomenon confirmed our previous assumption that an increase in the mixing entropy of the system will lead to a lower $\Delta G$ and improved stability. Our theoretical simulations (Supplementary Fig. 25) revealed surface energies of 0.22, 0.21, 0.19, and 0.18 eV per atom for Cu, $Cu_{97}Pd_3$, $Cu_{95}Sb_5$, and $Cu_{92}Sb_5Pd_3$, respectively, further confirming the improved stability of the $Cu_{92}Sb_5Pd_3$ SAA catalyst by co-doping Sb and Pd on a Cu base.

To gain a better understanding of the $CO_2$-to-CO pathway, we conducted in situ Raman spectroscopy, a sensitive technique for detecting CO$^*$ intermediates[19], to monitor the evolution of reactive intermediates. Figure 3a, b show the in situ Raman spectra acquired for four samples during a negative-going potential sweep from 0 to $-1.2$ V *vs*. RHE. Upon applying cathodic potentials, noticeable Raman peaks emerged from 2000 to 2100 cm$^{-1}$ for the three control samples. The high-frequency bands appearing at $\sim$2080 cm$^{-1}$ were attributed to CO$^*$ on step sites of the Cu base, whereas the low-frequency bands at $\sim$2045 cm$^{-1}$ correspond to CO$^*$ on terrace sites[20]. The emergence of two peak positions indicated a higher surface coverage of absorbed $^*$CO on the control samples. We also observed redshifts of these peaks at more negative potentials due to the Stark tuning effect for those three samples. In contrast, on $Cu_{92}Sb_5Pd_3$, only a weak peak appeared

at $\sim$2080 cm$^{-1}$ under a relatively negative potential, indicating a lower coverage of CO$^*$ intermediates. Moreover, we detected a peak at $\sim$360 cm$^{-1}$ associated with Cu-CO stretching[21,22], which was more pronounced on the three control samples than on $Cu_{92}Sb_5Pd_3$. These results clearly demonstrate a higher concentration of CO$^*$ intermediates on Cu, $Cu_{95}Sb_5$ and $Cu_{97}Pd_3$, which reasonably explains their higher productivities toward $C_{2+}$ products such as $C_2H_4$. We further explored the adsorption behavior of the chemical intermediates at the active sites by performing in situ attenuated total reflection surface-enhanced infrared absorption spectroscopy (ATR-SEIRAS) from 0 to $-1.0$ V *vs*. RHE. The in situ ATR-SEIRAS spectra in Supplementary Fig. 26 show similar bands related to surface-bond CO$^*$ at 2000-2100 cm$^{-1}$ among the four samples[23]. With a negatively sweeping potential, all CO$^*$ band frequencies redshifted due to the Stark effect[24]. Notably, at $-1.0$ V *vs*. RHE, the CO$^*$ peak almost vanished on $Cu_{92}Sb_5Pd_3$ but still remained on other samples, indicating an easier desorption of CO$^*$ intermediates from the $Cu_{92}Sb_5Pd_3$ catalyst surface to form gaseous CO[25].

To complement the in situ spectroscopic evidence from Raman and ATR-SEIRAS spectra, we also employed CO-diffuse reflectance infrared Fourier transform spectroscopy (CO-DRIFTS) measurements to support our results. Generally, the adsorption/desorption rate of CO$^*$ intermediates depends on their binding strength on different catalysts. A strong binding strength promotes CO$^*$-CO$^*$ coupling to $C_{2+}$ products[26], while facile binding inhibits the formation of coupling products. Hence, a slow adsorption/desorption rate enhances the possibility of coupling between CO$^*$ intermediates, leading to the generation of multicarbon products. Conversely, a fast adsorption/desorption rate favors the formation of gaseous CO since the coverage of CO$^*$ intermediates is low. To measure the desorption rate of CO$^*$ intermediates on four catalysts, we performed a series of CO-DRIFTS measurements. All samples were first exposed to gaseous CO until saturation and then swept with Ar to measure the desorption rates of preadsorbed CO ($CO_{ad}$). During the desorption process, the main peaks at $\sim$2100 cm$^{-1}$ in Fig. 3c were attributed to $CO_{ad}$ on Cu species[27–29], which reinforced the fact that Cu sites served as the absorption sites for CO in the four samples, consistent with the in situ spectroscopy results. After normalizing by peak area to the same range, Fig. 3d shows that the desorption rates of $CO_{ad}$ among the four samples rank as follows: $r_{CO\ (Cu92Sb5Pd3)} > r_{CO\ (Cu95Sb5)} > r_{CO\ (Cu97Pd3)} > r_{CO\ (Cu)}$. Therefore, it is rationalized that CO$^*$ intermediates are most likely and easiest to desorb on $Cu_{92}Sb_5Pd_3$ compared to the other three, which explains the near-unity selectivity of $Cu_{92}Sb_5Pd_3$ toward CO rather than $C_{2+}$ products.

In this work, we sought to coordinately tune the electronic structure of Cu by alloying two distinct single atoms, steering it towards selective CO production with enhanced activity and stability. To corroborate our hypothesis with direct experimental evidence, we probed the electronic structure of the catalysts by synchrotron valance band spectra (SVBS) measurements[30,31], as shown in Fig. 3e. These spectra reflect the density of state (DOS)[32] and showed that the 3$d$ bands of Cu in different samples varied with the composition of different single-atom metals. After adding Pd and Sb to the Cu base individually, the $d$-band center of Cu shifted downwards from 2.73 eV to 2.80 eV and 2.83 eV, respectively. Upon adding both single-atom metals simultaneously, the $d$-band center further shifted to 2.87 eV. This trend of the $d$-band centers indicated a logical change in the electronic structure, as we predesigned, which was also in accordance with our follow-up density functional theory (DFT) calculated deductions (Supplementary Fig. 27). It is generally accepted that the variation in $d$-band centers correlates with different adsorption energies for intermediates during the $CO_2$RR[4,5,9]. The lowest $d$-band center of $Cu_{92}Sb_5Pd_3$ forecasted a fairly weak binding strength of CO$^*$ intermediates to the catalyst surface, which facilitated their desorption to form gaseous CO.

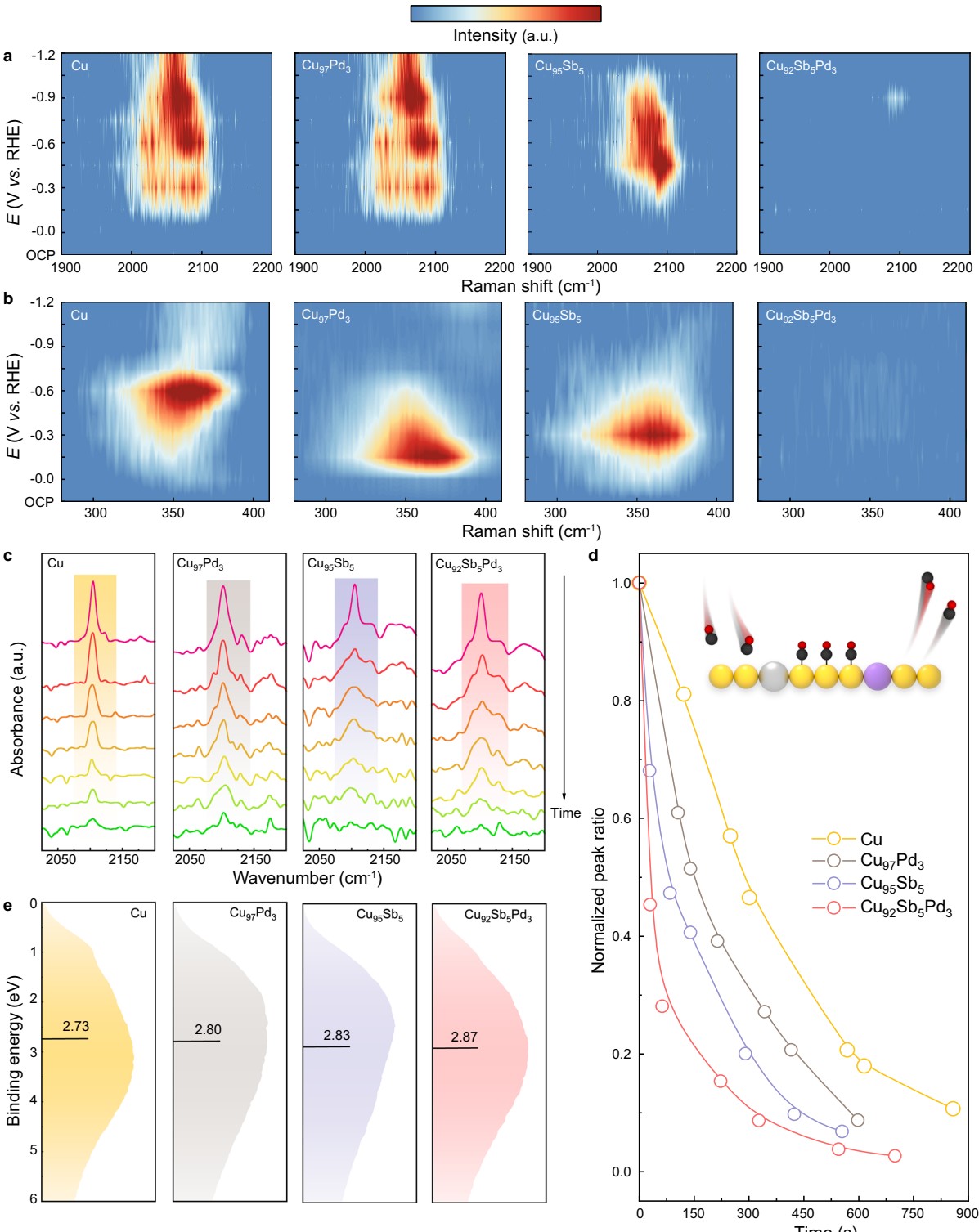

**Fig. 3 | Mechanistic studies of the electrochemical CO₂-to-CO conversion on Cu₉₂Sb₅Pd₃. a, b** In situ Raman spectra of four different catalysts at various potentials (reference to RHE). **c** CO-DRIFTS measurements of four different catalysts. **d** Normalized CO peak ratio obtained from **c** as a function of time. An illustration of the CO-DRIFTS mechanism is inserted at the top. **e** SVBS measurements of four as-synthesized catalysts.

To gain further insights into the origin of the stunning CO evolution performance of $Cu_{92}Sb_5Pd_3$, we performed theoretical simulations to investigate the effect of Pd and Sb dopants. As discussed previously[33], Cu (111) surface is more active for the $CO_2RR$ to CO than Cu (111) and Cu (100). For copper-based single-atom alloy catalysts, their simulation performance on step surface is in good agreement

with experimental results[4,5]. Thus, the Cu (211) surface model was finally chosen. Three models with different Pd doping positions were constructed, namely, $Cu_{92}Sb_5Pd_3$ (211), $Cu_{92}Sb_5Pd_3$ (211)−1 and $Cu_{92}Sb_5Pd_3$ (211)−2 (Supplementary Fig. 28). As shown in Fig. 4a, the adsorption energies of CO* on $Cu_{92}Sb_5Pd_3$ (211)−1 and $Cu_{92}Sb_5Pd_3$ (211)−2 are weaker than those on Cu (211), $Cu_{97}Pd_3$ (211) and $Cu_{95}Sb_5$ (211),

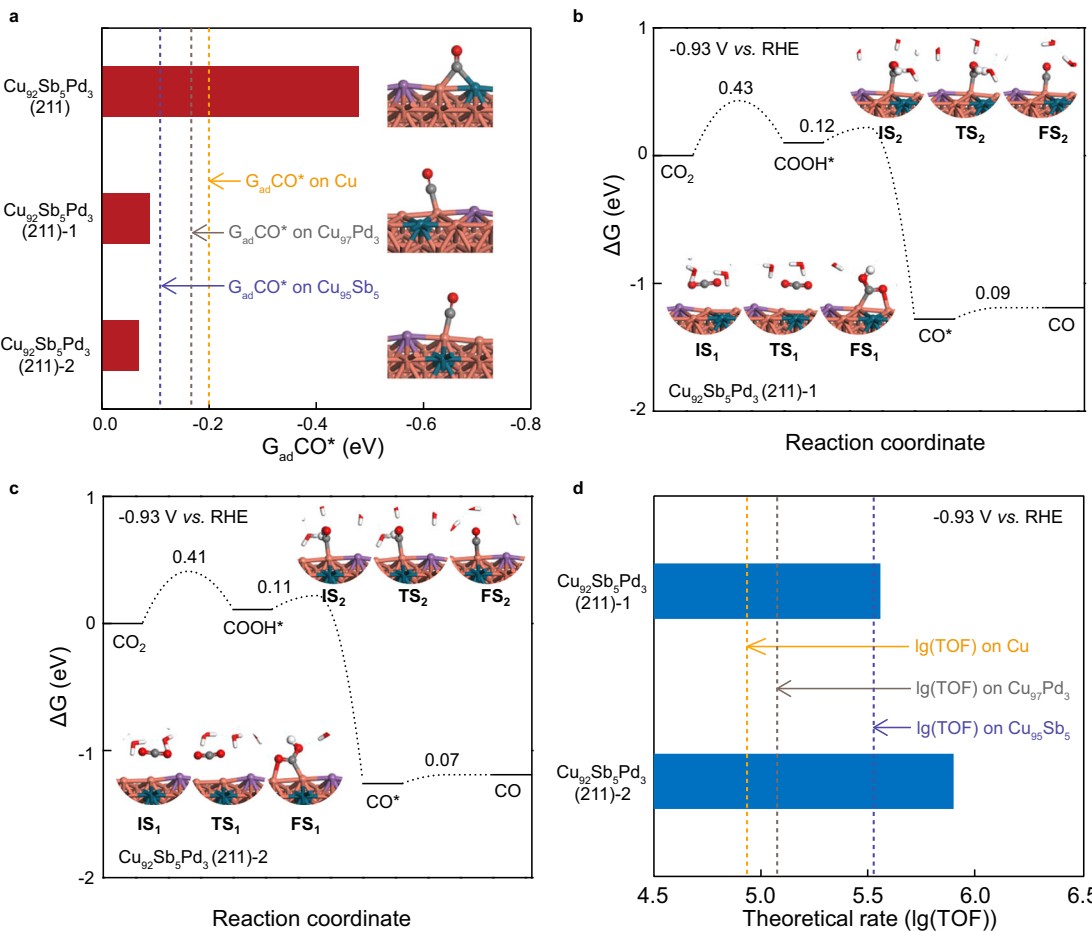

**Fig. 4 | DFT calculations. a** The calculated adsorption energy and structures of CO* on $Cu_{92}Sb_5Pd_3$ (211), $Cu_{92}Sb_5Pd_3$ (211)−1 and $Cu_{92}Sb_5Pd_3$ (211)−2, where the dashed lines refer to the adsorption energy of CO* on Cu (211) (−0.20 eV), $Cu_{97}Pd_3$ (211) (−0.18 eV) and $Cu_{95}Sb_5$ (211) (−0.11 eV). $CO_2RR$ to CO on $Cu_{92}Sb_5Pd_3$ (211)−1 (**b**) and $Cu_{92}Sb_5Pd_3$ (211)−2 (**c**). The initial (IS), transition (TS), and final (FS) structures are shown as insets, where Cu, Sb, Pd, C, O, and H are represented in orange, purple, green, gray, red, and white, respectively. The symbols with the same color represent the same atoms in figures **a**–**c**. **d** Theoretical rates of CO production on $Cu_{92}Sb_5Pd_3$ (211)−1 and $Cu_{92}Sb_5Pd_3$ (211)−2, where the dashed lines refer to lg(TOF) on Cu (211) (4.89), $Cu_{97}Pd_3$ (211) (5.09) and $Cu_{95}Sb_5$ (211) (5.54).

where CO* was adsorbed at the top site on the Cu atom adjacent to the Sb and Pd atoms. Similarly, the $G_{ad}$ CO* on Cu (211), $Cu_{97}Pd_3$ (211) and $Cu_{95}Sb_5$ (211) were all obtained with CO* adsorbed at the top sites on Cu atoms (Supplementary Fig. 29). However, when CO* was adsorbed at the bridge site between Cu and Pd atoms on $Cu_{92}Sb_5Pd_3$ (211), the adsorption energy of CO* on $Cu_{92}Sb_5Pd_3$ (211) was stronger than that on Cu (211), $Cu_{97}Pd_3$ (211) or $Cu_{95}Sb_5$ (211). Based on the experimental characterization results, where weakened CO* adsorption was found on $Cu_{92}Sb_5Pd_3$ relative to either Cu or $Cu_{95}Sb_5$, the bridge Cu site on $Cu_{92}Sb_5Pd_3$ (211) should not be the predominant active site. Hence, $Cu_{92}Sb_5Pd_3$ (211)−1 and $Cu_{92}Sb_5Pd_3$ (211)−2 were chosen for the subsequent calculation and analysis. To further excavate the difference between $Cu_{92}Sb_5Pd_3$ (211)−1 and $Cu_{92}Sb_5Pd_3$ (211)−2, the electrochemical barriers of the $CO_2RR$ to CO over these two structures were calculated. The electrochemical barriers were first calculated on the basis of the "charge extrapolation" method[34,35] within the capacitor model[36]. The amount of electron transfer ($\Delta q$) from the water layer to the electrode is linearly correlated with the relative work function ($\Phi$) at the initial states (IS), transition states (TS), and final states (FS) (Supplementary Figs. 30–33). We chose a cathodic potential of −0.93 V $vs$. RHE for further theoretical simulations, at which the highest $FE_{CO}$ of 100% (±1.5%) could be reached at −402 mA cm⁻² on $Cu_{92}Sb_5Pd_3$. Figure 4b, c show that at −0.93 V $vs$. RHE, the kinetic barrier of CO formation on $Cu_{92}Sb_5Pd_3$ (211)−2 is lower than that on $Cu_{92}Sb_5Pd_3$ (211)−1. In addition, $Cu_{92}Sb_5Pd_3$ (211)−2 shows lower barriers for the

hydrogenation of $CO_2$ and COOH* than Cu (211), $Cu_{97}Pd_3$ (211) and $Cu_{95}Sb_5$ (211) (Supplementary Fig. 34). Besides, microkinetic modelling over $Cu_{92}Sb_5Pd_3$ (211)−1 and $Cu_{92}Sb_5Pd_3$ (211)−2 was also conducted at −0.93 V $vs$. RHE (Fig. 4d). The theoretical rates of CO production on $Cu_{92}Sb_5Pd_3$ (211)−1 and $Cu_{92}Sb_5Pd_3$ (211)−2 are both higher than those on Cu (211), $Cu_{97}Pd_3$ (211) and $Cu_{95}Sb_5$ (211), among which $Cu_{92}Sb_5Pd_3$ (211)−2 shows the highest theoretical activity. Taken together, $Cu_{92}Sb_5Pd_3$ (211)−2 is supposed to be the major active structure, while $Cu_{92}Sb_5Pd_3$ (211)−1 tends to be suboptimal. The barriers and reaction free energies of the main and side reactions are summarized in Supplementary Tables 6 and 7. The TOFs of the different products for $CO_2RR$ and HER on $Cu_{97}Pd_3$ (211), $Cu_{95}Sb_5$ (211), $Cu_{92}Sb_5Pd_3$ (211)−1 and $Cu_{92}Sb_5Pd_3$ (211)−2 surfaces at −0.93 V $vs$. RHE were additionally calculated and listed in Supplementary Table 8. As shown in Supplementary Fig. 35, the calculated $FE_{CO}$ follow the order of $Cu_{97}Pd_3$ (211) < $Cu_{95}Sb_5$ (211) < $Cu_{92}Sb_5Pd_3$ (211)−2, which is comparable to the experimental results for all three catalysts. At the steady state, the CO* coverages for $CO_2RR$ on different models follow the order of $Cu_{92}Sb_5Pd_3$ (211)−2 (1.5%) <$Cu_{92}Sb_5Pd_3$ (211)−1 (3%) <$Cu_{95}Sb_5$ (211) (7%) <$Cu_{97}Pd_3$ (211) (51%) <Cu (211) (70%) (Supplementary Fig. 36), which is consistent with the above in situ Raman measurements. As such, reasonably, the $Cu_{92}Sb_5Pd_3$ catalyst shows the highest $CO_2RR$ activity towards exclusive CO production compared with the bimetallic counterparts or pristine Cu. Beyond the above, to investigate the charge redistribution between Sb/Pd additions and the Cu matrix,

Bader charge analysis was also conducted. As shown in Supplementary Fig. 37, on $Cu_{92}Sb_5Pd_3$, the copper atoms present partial electron-deficient states, while Sb/Pd atoms express an electron-rich feature, further attesting to the former *operando* XAS analysis.

Overall, we showcase an enlightening design principle for creating trimetallic SAAs by alloying Cu with two distinct single-atom metals for the selective $CO_2RR$ to CO. Both experimental and theoretical results validate the effectiveness of our design strategy. The synergistic effects of both Sb and Pd single atoms on Cu not only modulate the electronic structure of Cu to favor CO formation and inhibit the HER but also enhance the stability of the catalyst. As a result, the $Cu_{92}Sb_5Pd_3$ catalyst exhibits outstanding performance in $CO_2$-to-CO conversion, achieving extremely high current density, near-unity selectivity and robust durability, outperforming many noble metal catalysts. In a broader context, our concept demonstrated here may be further extended to other element combinations and various electro-catalytic reactions.

## Methods

### Chemicals
Copper (II) chloride ($CuCl_2$, 98%) and palladium nitrate dihydrate ($Pd(NO_3)_2 \cdot 2H_2O$, 99.95%) were purchased from Aladdin. Antimony trichloride ($SbCl_3$, 99.9%), sodium borohydride ($NaBH_4$, 97%), ethanol (EtOH, 99.7%) and isopropanol (IPA, 99.5%) were purchased from Macklin. All chemicals were used without further purification.

### Synthesis
Generally, the $Cu_{92}Sb_5Pd_3$ catalyst was synthesized by modifying a previously reported method using $NaBH_4$ to reduce $CuCl_2$ and $SbCl_3$ precursors[10,11]. The details are as follows. First, solution A was prepared by dissolving 14 mg of $Pd(NO_3)_2 \cdot 2H_2O$, 28 mg of $SbCl_3$ and 294 mg of $CuCl_2$ in pure EtOH. Later, the mixture was sonicated for 15 min to obtain a clear solution. On the other hand, 0.95 g of $NaBH_4$ was dissolved in 11 mL of EtOH/water at a volume ratio of 8:3 at 4 °C to prepare solution B. Afterwards, solution A was rapidly added to solution B in a beaker at 4 °C with continuous stirring for 1 h under an Ar atmosphere. Note that the beaker should be tightly sealed with parafilm to isolate the air to prevent further oxidation. After a violent reaction, the obtained black precipitate was then washed with DI water and IPA three times and dried under vacuum at room temperature for 8 h. The above cleaning procedures were finished in a few minutes to prevent oxidation in air. The obtained samples were then stored in a glove box under an Ar atmosphere.

For the synthesis of other controlled samples, the solutes in solution A were changed to 14 mg of $Pd(NO_3)_2 \cdot 2H_2O$ and 228 mg of $CuCl_2$ (in the case of the $Cu_{97}Pd_3$ sample), 28 mg of $SbCl_3$ and 314 mg of $CuCl_2$ (in the case of the $Cu_{95}Sb_5$ sample), and 300 mg of $CuCl_2$ (in the case of the Cu sample). The following steps were the same as those for the $Cu_{92}Sb_5Pd_3$ catalyst.

### Electrochemical measurements
**$CO_2RR$ performance test.** All the electrochemical measurements were conducted at room temperature using BioLogic VMP3 and CHI (1140c). Typical three-electrode cell measurements were performed using a conventional flow cell. To prepare the cathode electrode, precursor ink (12 mg of catalyst mixed with 24 μL of 5% Nafion 117 solution dissolved in 2 mL of IPA) was spray-coated onto a gas diffusion layer (YLS-30T) with a mass loading of ~1 mg cm$^{-2}$ using an air brush, and eventually air-dried on a hotplate at 60 °C. A Ag/AgCl wire in a saturated KCl solution was used as the reference electrode, and Ni foam was used as the counter electrode. The working and counter electrodes were then placed on opposite sides of two 1 cm-thick polytetrafluoroethylene (PTFE) sheets with 0.4 cm × 1.5 cm channels such that the catalyst layer interfaced with the flowing electrolyte. The geometric surface area of the catalyst was 0.6 cm$^2$. A Nafion 115 membrane (Fuel Cell Store) was sandwiched between the two PTFE sheets to separate the chambers. A schematic illustration of the flow-cell configuration is provided in Supplementary Fig. 38. $CO_2$ flowed through the gas room behind the cathode, and the flow rate was maintained at 30 sccm (monitored by an Alicat Scientific mass flow controller). In addition, 0.5 M $KHCO_3$ (pH = 7.4, or 7.2 if $CO_2$ saturated) was circulated as the cathode electrolyte at a flow rate of 1.1 mL min$^{-1}$, while 1 M KOH was purged as the anode electrolyte. All potentials were converted to the RHE reference scale using the relation $E_{RHE} = E_{Ag/AgCl} + 0.197 + pH \times 0.0592 - 85\% \times i \times R$, where $R$ is the solution resistance and the compensation coefficient is taken as 85% for $iR$ compensation during flow cell operation. All potentials measured using the three-electrode set-up were manually compensated by $iR$ correction. At least three independent measurements were carried out under each current.

**$CO_2RR$ product analysis.** The gaseous products were tested by online gas chromatography (GC) (PerkinElmer Clarus 690), which was equipped with a flame ionization detector, a thermal conductivity detector, and a Molsieve 5 Å column. The liquid products were quantified by a 400 MHz nuclear magnetic resonance (NMR) spectrometer (BUKER) and ion chromatography (IC) (Thermo Fisher Scientific ICS-600). For NMR tests, 100 μL of $D_2O$ (Sigma Aldrich, 99.9 %) and 0.05 μL of dimethyl sulfoxide (DMSO) (Sigma Aldrich, 99.9%) as an internal standard were added to 600 μL of the electrolyte after electrolysis.

**ECSA and $H_2$ desorption measurements.** Both experiments were performed using a customized gas-tight H-type glass cell with 0.5 M $KHCO_3$ (pH = 7.4) as the electrolyte. The anode and cathode were separated by a Nafion 115 film (Fuel Cell Store). The catalysts were loaded on glassy carbon electrodes. Before the experiments, the cathode electrolyte was bubbled with Ar for at least 20 min to remove $CO_2$. In ECSA measurements, the electrical double-layer capacitance ($C_{dl}$) is calculated by plotting the relationship of $\Delta j = ja-jc$, where $ja$ and $jc$ are the positive and negative scan currents, respectively. And ECSA = $C_{dl} / C_{dl-ref}$, where $C_{dl}$ is derived from the slope of the $\Delta j$ as a function of the scan rate and the number of $C_{dl-ref}$ is taken as 29[37].

**Tafel plot.** Tafel plots were generated to evaluate the catalytic kinetics of the $CO_2RR$ and fitted with the following equation: $\eta = k \times lg(j_{CO}) + b$, where $j_{CO}$ is the CO partial current density and $\eta$ is the overpotential for $CO_2 + H_2O + 2e^- \rightarrow CO + 2OH^-$ ($E^0 = -0.11$ V vs. RHE). A smaller slope k indicates faster kinetics for CO production. If the rate-determining step (RDS) is the first electron transfer from $CO_2$-to-$^*CO_2^-$, the Tafel slope is calculated by the following formula:

$$\frac{\partial(-\eta)}{\partial lg(j_{CO})} = \frac{2.3RT}{\alpha F} \tag{1}$$

In this equation, $\alpha$ is the transfer coefficient and $F$ is the Faraday constant. The standard values of the Tafel slopes based on different RDSs are further given in Supplementary Table 4.

**In situ DEMS measurements.** In situ differential electrochemical mass spectrometry (DEMS) was performed using a custom-made electrochemical capillary DEMS flow cell. The catalysts were loaded on the gas diffusion layer (GDL) as the cathode, where $CO_2$ flowed behind. In addition, 0.5 M $KHCO_3$ (pH = 7.4, or 7.2 if $CO_2$ saturated) was used as the electrolyte. A capillary was put into the gas outlet of the flow cell to draw the gas products into the DEMS sensor (PrismaPro). The signals at mass-to-charge ratios (m/z) of 26 and 28 represent $C_2H_4$ and CO, respectively. Linear sweep voltammetry (LSV) with a scan rate of 5 mV s$^{-1}$ was conducted on the cathode. The onset potentials were determined according to the positions where the signal-to-noise ratio was greater than 5.

**Long-term stability test.** A membrane electrode assembly (MEA) was used for the long-term stability test with a zero-gap configuration where the anode, membrane, and cathode were compressed together to form one reactor. The cathodic electrode area was 4 cm². An IrO₂/Ti mesh was used as the anode, and an anion exchange membrane (Sustainion X37-50 Grade 60, Dioxide Materials) was placed between the anode and cathode. A schematic illustration of the MEA is provided in Supplementary Fig. 39. $CO_2$ was directly fed to the GDL cathode at 40 sccm (monitored by an Alicat Scientific mass flow controller). Additionally, 0.1 M $KHCO_3$ was purged as the anode electrolyte, which was replaced every four days during the 22-day-long stability test.

**Characterization techniques.** Transmission electron microscopy (TEM) images and energy dispersive X-ray (EDX) elemental mapping were obtained on a Tecnai G2 F20 S-TWIN using Mo-based TEM grids. High-angle annular dark-field scanning transmission electron microscopy (HAADF-STEM) images and corresponding energy-dispersive spectra (EDS) elemental mapping were measured on a JEOL ARM-200F field-emission transmission electron microscope operated at 200 kV using Mo-based TEM grids. X-ray diffraction (XRD) patterns were recorded using a Shimadzu X-ray diffractometer (XRD-6100, Japan) with Cu-Kα radiation (λ = 1.54178 Å). X-ray photoelectron spectroscopy (XPS) measurements were performed on a Kratos-Axis Supra XPS spectrometer with an excitation source of Al Kα = 1486.6 eV. The binding energies obtained in the XPS spectral analysis were corrected by referencing C 1$s$ to 284.6 eV. The X-ray absorption spectra (XAS) of the Cu $K$-edges, Sb $K$-edges and Pd $K$-edges were obtained at BL14W1 beamlines at the Shanghai Synchrotron Radiation Facility (SSRF) under "top-up" mode with a constant current of 200 mA and recorded under fluorescence mode in an H-cell with a Lytle detector. The spectra were processed and analysed by the software codes Athena and Artemis. In situ Raman analysis was performed using a Renishaw inVia Raman analyser equipped with a 785 nm laser combined with a custom flow cell. During the experiments, the laser was focused on the surface of the sample with a laser intensity of 1 mW. In situ electrochemical attenuated total reflection surface-enhanced infrared absorption spectroscopy (ATR-SEIRAS) and CO-diffuse reflectance infrared Fourier transform spectroscopy (CO-DRIFTS) were conducted on a Thermo Scientific Nicolet iS50 FTIR spectrometer at room temperature. Si crystals were used in the ATR-SEIRAS experiments. Before the experiments, a polycrystalline Au film was deposited onto the Si crystal *via* chemical bath deposition. Typically, the polished Si crystal was first immersed in an $NH_4F$ bath for 2 minutes. Au plating solutions with 5.75 mM $NaAuCl_4 \cdot 2H_2O$, 0.025 M $NH_4Cl$, 0.025 M $Na_2S_2O_3 \cdot 5H_2O$ (98%), 0.075 M $Na_2SO_3$ (98%), and 0.026 M NaOH (99.99%) were prepared. Then, 0.8 mL of 2 wt% HF aqueous solution was mixed with 4.4 mL of the above Au plating solution, after which the Si surface was immersed in the above mixed solution for 15 min at 55 °C and later rinsed with water. After the above preparation process, the precursor ink was spray-coated onto the Au film by an air brush for further use. For CO-DRIFTS measurements, ZnSe and an incident light window were used to examine highly scattering powder samples in diffuse reflectance mode[38]. Synchrotron valance band spectra (SVBS) measurements were performed at BL10B of the National Synchrotron Radiation Laboratory (NSRL) using synchrotron-radiation light as the excitation source with a photon energy of 100 eV.

**Computational details.** Density functional theory (DFT) calculations were performed by the Vienna ab initio simulation package (VASP)[39,40]. The generalized gradient approximation (GGA) of the revised Perdew-Burke-Ernzerhof (rPBE) functional[41] was used. We chose the projected augmented wave (PAW) method[42,43] and a plane wave basis set with a kinetic energy cutoff of 400 eV. Geometry optimizations were performed with a force convergence smaller than 0.05 eV Å⁻¹. All surface models were built with four layers comprising 48 atoms. The two layers at the bottom were fixed, while the other atoms relaxed. A Monkhorst-Pack $k$-point of (4 × 2 × 1) was used for the optimization of all surface structures.

The adsorption energies of intermediates were referenced to the gas phase energies of CO, $H_2O$, and $H_2$. The reaction-free energies ($\Delta G$) were calculated as follows: $\Delta G = \Delta E + \Delta ZPE - T\Delta S$ ($T$ = 300 K), where $\Delta E$ is the electronic energy based on DFT calculations directly, and $\Delta ZPE$ and $\Delta S$ are the corrections of the zero point energy and entropy, respectively. The climbing image nudged elastic band (CI-NEB) method was used to locate the transition states[44]. The solvation effect was also calculated using implicit models through VASPsol calculation[45]. In addition, the chemical potential of ($H^+ + e^-$) was calculated by G ($H^+ + e^-$) = ½ G ($H_2$) at 0 V *vs*. RHE. A computational hydrogen electrode model was used to calculate the free energy change at varying potential[46].

The surface energy (γ) is calculated with the following formula:

$$\gamma = \frac{E_{slab} - \sum n_i \mu_i}{N} \qquad (2)$$

$E_{slab}$ is the energy of the slab model. $\mu_i$ is the energy of an $i$ atom in the bulk, and $n_i$ is the number of $i$ atoms in the slab model ($i$ = Cu, Sb and Pd). $N$ is the number of all atoms in the slab model.

The electrochemical barriers ($G_a$) were calculated on the basis of the "charge-extrapolation" method[34] within the capacitor model. The amount of electron transfer ($\Delta q$) from the water layer to the electrode is linearly correlated with the relative work function ($\Phi$) at the initial state (IS), transition state (TS), and final state (FS). According to the capacitor model, the energy change between two states at a constant work function can be calculated as follows:

$$E_2(\Phi_1) - E_1(\Phi_1) = E_2(\Phi_2) - E_1(\Phi_1) + \frac{(q_2 - q_1)(\Phi_2 - \Phi_1)}{2} \qquad (3)$$

$$E_2(\Phi_2) - E_1(\Phi_2) = E_2(\Phi_2) - E_1(\Phi_1) - \frac{(q_2 - q_1)(\Phi_2 - \Phi_1)}{2} \qquad (4)$$

where $E_1(\Phi_1)$ and $E_2(\Phi_2)$ correspond to the energies of states 1 and 2, respectively. $\Phi$ and q refer to the work function and interfacial charge transfer, respectively.

Setting $\Delta E(\Phi) = E_2(\Phi) - E_1(\Phi)$ at a given work function $\Phi$ and $\Delta q = q_2 - q_1$, the following equation can be derived:

$$\Delta E(\Phi_2) - \Delta E(\Phi_1) = -\Delta q(\Phi_2 - \Phi_1) \qquad (5)$$

where $\Delta E(\Phi_1)$ and $\Delta E(\Phi_2)$ are the barriers at $\Phi_1$ and $\Phi_2$, respectively. The work function $\Phi$ can be related to the absolute potential ($U_{SHE}$) by $U_{SHE} = \frac{\Phi - \Phi_{SHE}}{e}$, where $\Phi_{SHE}$ has been determined experimentally to be ~ 4.4 eV. Therefore, the potential-dependent barrier can be calculated by this method.

Microkinetic modelling was used to simulate the reaction rate in the $CO_2RR$ and HER, which was solved by the CATKINAS code[47,48]:

$$\frac{\partial \theta_i}{\partial t} = 0 \qquad (6)$$

$$\sum_i \theta_i = 1 \qquad (7)$$

The reaction rate on surfaces was described by ref. 49

$$r = \theta_A \theta_B \frac{k_B T}{h} e^{-G_a/k_B T} \qquad (8)$$

The reaction rate of the $CO_2RR$ to $C_{2+}$ products was estimated based on the Arrhenius equation and the CO* coverage at steady state

The FE was described by the following equation:

$$FE/\% = \frac{n(i)\mathrm{TOF}(i)}{\sum n(i)\mathrm{TOF}(i)} \times 100 \qquad (9)$$

where n($i$) represents the electron transfer number and TOF($i$) is the turnover frequency obtained by microkinetic simulation for product $i$.

## Data availability

All experimental data are available in the main text or the supplementary materials. Source data of the figures in the main text are provided. Source data are provided with this paper.

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

## Acknowledgements

C.X. acknowledges the National Key Research and Development Program of China (2022YFB4102000) and NSFC (22102018 and 52171201). J.X. acknowledges the National Natural Science Foundation of China (No. 22321002), the Energy Revolution S&T Program of Yulin Innovation Institute of Clean Energy (Grant No. YIICE E411050316), Strategic Priority Research Program of the Chinese Academy of Sciences (No. XDB36030200) and the DICP (DICP I202314 and DICP I202425). T.Z. acknowledges the NSFC (22322201 and 22278067), and the Natural Science Foundation of Sichuan Province (2023NSFSC0094). We thank beamline BL14W1 of Shanghai Synchrotron Radiation Facility and BL10B of National Synchrotron Radiation Laboratory for providing the facilities.

## Author contributions

This project was conceptualized by C.X. and was supervised by C.X. and J.X. Jing.X. conducted all the experiments with the help of the following co-authors. C.L., Y.D. and L.L. helped with the stability test. J.L. helped with DEMS measurements. W.X. helped with Raman analysis. Y.J. provided the illustration of the CO-DRIFTS mechanism. X.Z. provided useful discussion on this work. X.L., Q.J. and T.Z. offered guidance in writing the paper. X.D. and J.X. performed the DFT calculations. Jing.X., C.X., and J.X. wrote the paper with input from all authors. All authors discussed the results and commented on the manuscript.

## Competing interests
