## [Peer Review File · Nature Communications]

REVIEWER COMMENTS

Reviewer #1 (Remarks to the Author):

In this work, trimetallic single-atom alloy (SAA) catalyst Cu₉₂Sb₅Pd₃ is synthesized, which shows 85% CO selectivity at -1 A cm⁻² and 528 hours stability at -100 mA cm⁻². Doping of Sb improves CO generation and the adding of Pd is favorable for the maintaining of SAA structure. Electrochemical reduction of CO₂ to CO with high selectivity has been repeatedly reported on many metallic and nonmetallic catalysts in the last few years, so the catalytic performance doesn't appear to be the state-of-the-art. The novelty in mechanistic understanding is also lacking, and thus the reviewer cannot recommend the publication of this work in the Nature communications.

Major comments/questions:

1. What are the structure motif with high CO selectivity? Since the amounts of Sb and Pd are relatively small compared to Cu, how come they are able to change the properties of all Cu sites? Judging from the TEM images, the surface coverage of dopants is quite low.
2. FE of CO at -1 A cm⁻² is 85% in the main text, and therefore the claim of “~100% CO selectivity” in the abstract could be misleading.
3. Although the performance of Cu₉₂Sb₅Pd₃ surpasses most of Au and Ag catalysts, noble metal Pd is used in this catalyst makes its .
4. It is well established that catalysts with weak *CO adsorption energy is favorable for CO₂ to CO in the experimental and computation literature. The authors devoted much of the characterization results (Fig.3) and discussion to re-prove this point, which doesn't yield much new insights.
5. Pd possesses stronger *CO binding energy than Cu. Why the doping of Pd leads to the increase of desorption rates of CO_{ad} in Fig.3d?

Reviewer #2 (Remarks to the Author):

In this work, the authors have synthesized a trimetallic single atom alloy catalyst (Cu₉₂Sb₅Pd₃) for the electrochemical reduction of CO₂ (CO₂RR). This Cu-based catalyst has shown an impressive CO selectivity (~100%) and a high Faradic efficiency (95%). The authors have investigated the roles of Sb and Pd single atoms via a series of spectroscopic techniques and theoretical simulations. It has been found that the presence of Sb and Pd atoms not only improved the stability of the alloy but also

increased the selectivity of CO by decreasing the CO binding affinity on Cu atoms. While qualitative agreements between theoretical simulations and experimental measurements have been achieved, I believe that it would be more beneficial to the identification and understanding of the nature of the active center if more quantitative comparisons could be provided. In my opinion, this study is interesting and suitable for publication in Nature Communication after revision. Below are some comments:

1. A key concern is the spatial distribution of Sb and Pd single atoms, in particular, their preferences for the surface or the subsurface. Providing evidence or discussion on this aspect would enhance the clarity of the catalyst's structures.

2. As compared to the other samples, the high CO-selectivity of the Cu₉₂Sb₅Pd₃ catalyst has been attributed to its weaker CO binding affinity, which strongly depends on the active site model where the CO is adsorbed. The structure of the active site, however, is solely determined by the comparison of calculated CO adsorption energies on three different models with other samples (Cu, Cu₉₇Pd₃, and Cu₉₅Sb₅). Is there any direct correspondence between the theory and experiment for the structure of the active site (e.g. from the spectroscopic evidence)?

3. The authors have investigated three types of trimetallic active structures by DFT calculations. According to the CO binding affinity and the activity, the authors suggested the most active model Cu₉₂Sb₅Pd₃(211)-2 as the active site, while the Cu₉₂Sb₅Pd₃(211) model is excluded because of its stronger CO binding affinity as compared to Cu. Does the existence of other types of trimetallic active structures (e.g. Cu₉₂Sb₅Pd₃(211)) influence the reaction rate of side reactions and the selectivity? The authors should provide more explanations on why Cu(211) was employed in the modeling.

4. I recommend the authors to provide more discussions on the individual roles of Sb and Pd in modifying catalytic performance and the synergistic cooperation between them.

5. While the TOFs of different samples/models have been calculated via theoretical simulations, they cannot be compared to experimental measurements directly. It would be more convincing if the authors calculate and compare the TOFs of side reactions with the TOF of the major reaction, such that the selectivity can be obtained and compared with experimental measurements.

6. In this manuscript (line 83), the mix entropy was emphasized. On the other hand, providing more illustrations regarding the contributions of enthalpy, particularly interactions among metals are desirable, as they often play a significant role in determining free energy changes.

7. The authors should provide more details of “charge extrapolation” method within the capacitor model. Also providing the atomic coordinates of DFT calculations is encouraged.

Reviewer #3 (Remarks to the Author):

This work presents an intriguing design principle for synthesizing a trimetallic catalyst via single-atom alloying strategy to achieve CO₂-to-CO conversion with excellent selectivity, activity, and stability. In their previous work (Nat. Commun. 14, 340, 2023), the group presented a bimetallic Sb-alloyed copper catalyst for CO production. But its production rate and stability are still inferior to those of noble metals like Au and Ag. As a follow-up, in this current work, the incorporation of mixed entropy concepts into the trimetallic Cu-Sb-Pd system results in a holistic enhancement and overall upgrade in CO₂ conversion, almost on par with Ag/Au. The synergy of Pd/Sb single atoms significantly improves the selectivity and activity of the catalyst by adjusting the d-band center of Cu to an optimal state. Additionally, this approach reinforces the resistance of catalyst to deactivation, as the optimized surface energy elevates the migration energy of the sites. In this regard, this work marks a noteworthy achievement in both catalytical performance and mechanism exploration. This paper is well written and provides valuable insights for the community, which is a truly breakthrough. I recommend its acceptance after addressing the following issues.

(1) The authors showed that the control samples Cu₉₂Sb₈ and Cu₉₂Pd₈ have inferior selectivity toward CO compared to Cu₉₂Sb₅Pd₃ in Supplementary Fig.18. Given that the additional component contents are all equal to 8%, it would be helpful if the authors delve into the underlying reason for this observation. In particular, why does increasing the Pd content in Cu₉₂Pd₈ not suppress the C-C coupling on the Cu matrix? What is the role of the single-atom alloying strategy in tuning the selectivity of the catalyst?

(2) The authors use “mixing entropy” to measure the free energy of the system, the terminology of which resembles the entropy-based definition of high-entropy alloys (ACS Catal. 2020, 10, 11280-11306). Is there any connection between the Cu₉₂Sb₅Pd₃ catalyst and HEA? Or can it be categorized as HEA or MEA (medium-entropy alloy)?

(3) Although this work establishes the outstanding catalytic performance of the Cu₉₂Sb₅Pd₃ catalyst in CO₂ reduction, schematic illustrations of the cell configurations used in the experiments are notably absent. To enhance the clarity of the methodology, I recommend providing additional details on the cell structures employed.

(4) Considering the potential industrial applications of the Cu₉₂Sb₅Pd₃ catalyst, I recommend calculating cathodic energy efficiencies in a flow cell under applied potentials to further evaluate its viability for practical use.

(5) In Supplementary Fig.26, the scale bars among four catalysts vary from each other. Although they don't impact the analysis and conclusion of the ATR-SEIRAS results, it would be preferred if the authors elucidate the cause for the difference.

(6) As the electrode preparation process might result in partial oxidation of the catalyst, clear descriptions of this process should be given in Methods.

Chuan Xia	Mater. & Energy School
Professor of Chemistry	UESTC, China
Phone: +86 028-61838256	2006 Xiyuan Ave,
Email: chuan.xia@uestc.edu.cn	Chengdu, Sichuan 611731

Manuscript Number: **NCOMMS-23-57392**

Title: **“Turning copper into an efficient and stable CO evolution catalyst beyond noble metals”**

Authors: Jing Xue, Xue Dong, Chunxiao Liu, Jiawei Li, Yizhou Dai, Weiqing Xue, Laihao Luo, Yuan Ji, Xiao Zhang, Xu Li, Qiu Jiang, Tingting Zheng, Jianping Xiao, Chuan Xia

Corresponding authors: Jianping Xiao, Chuan Xia

Response to reviewers' comments:

We thank the editor and reviewers for their constructive comments, which have helped us to greatly improve our work and the quality of our manuscript. We have now performed substantial experiments and included additional analyses to fully address the reviewers' concerns and suggestions. Below, we address the points raised by the reviewers one by one in the following pages.

Reviewer 1

In this work, trimetallic single-atom alloy (SAA) catalyst Cu₉₂Sb₅Pd₃ is synthesized, which shows 85% CO selectivity at -1 A cm⁻² and 528 hours stability at -100 mA cm⁻². Doping of Sb improves CO generation and the adding of Pd is favorable for the maintaining of SAA structure. Electrochemical reduction of CO₂ to CO with high selectivity has been repeatedly reported on many metallic and nonmetallic catalysts in the last few years, so the catalytic performance doesn't appear to be the state-of-the-art. The novelty in mechanistic understanding is also lacking, and thus the reviewer cannot recommend the publication of this work in the Nature communications.

Response

We appreciate the reviewer's comments and the opportunity to improve our manuscript. However, we believe that the reviewer has overlooked some key aspects of our work that demonstrate its novelty and significance. We would like to address the reviewer's concerns and clarify our contributions as follows:

The reviewer suggested that our catalytic performance does not reach the state-of-the-art standard, but we contend that a fair comparison necessitates an understanding of our distinct objectives. Our work is not aimed at achieving the highest CO selectivity or stability among all catalysts but rather at enhancing the performance of Cu-based catalysts, which are more abundant and less expensive than noble metals. To the best of our knowledge, our Cu₉₂Sb₅Pd₃ SAA catalyst shows the highest CO selectivity and stability among Cu-based catalysts reported thus far, as well as comparable or better performance than most noble metal catalysts (see **Fig. 2f** in our manuscript for details).

Fig. 2f | Comparison of state-of-the-art noble metal catalysts with the $\text{Cu}_{92}\text{Sb}_5\text{Pd}_3$ catalyst in the CO_2RR .

To address the concern regarding the perceived lack of novelty in mechanistic understanding, we respectfully contest this notion. Our work provides new insights into the role of Sb and Pd single-atom doping in modulating the electronic structure and catalytic activity of Cu-based SAAs. We used a combination of experimental and theoretical methods to reveal how Sb and Pd doping affect CO_2 adsorption, activation, and reduction pathways on the Cu surface, as well as the HER suppression mechanism and improvement in stability. We also propose a general design principle for creating trimetallic SAAs with tunable properties for CO_2 reduction. These findings are novel and valuable for the field of CO_2 electrocatalysis and have not been previously reported in other studies.

Therefore, we respectfully request that the reviewer reconsider our work and acknowledge its originality and importance. We have revised our manuscript to highlight our contributions and address the reviewer's comments more clearly. We have also corrected any ambiguous or unclear expressions in our manuscript. We hope that the reviewer will find our revised manuscript suitable for publication in *Nature Communications*.

Comment 1

What are the structure motif with high CO selectivity? Since the amounts of Sb and Pd are relatively small compared to Cu, how come they are able to change the properties of all Cu sites? Judging from the TEM images, the surface coverage of dopants is quite low.

Response

We thank the reviewer for raising this point. We realize that the reviewer may have had some misunderstandings about our work, which may have led to doubts about the effectiveness of the small amounts of Sb and Pd dopants on the Cu matrix. Therefore, we would like to explain two main points of our work in more detail.

First, we would like to emphasize that the structural motif with high CO selectivity on our Cu₉₂Sb₅Pd₃ catalyst is not the entire Cu surface, but only the Cu atoms adjacent to both the Sb and Pd single atoms, as illustrated in **Fig. R1**. The co-introduction of two single-atom components collaboratively alters the electronic state of **neighbouring copper atoms**, making them more favourable for CO formation and less prone to the HER. This is supported by the DFT calculations in **Fig. 4** and **Supplementary Fig. 34** (copied below), which show that the Cu atoms affected by both the Sb and Pd single atoms in Cu₉₂Sb₅Pd₃ have lower energy barriers and higher theoretical rates for CO production than the unaltered Cu atoms in Cu (211). This accounts for their enhanced performance in CO₂-to-CO conversion compared to that of the unmodified Cu atoms.

Second, not all Cu atoms in Cu₉₂Sb₅Pd₃ are activated by the two single-atom dopants. Nevertheless, due to the higher kinetic barriers and poorer theoretical activities in the CO₂RR, the unaffected Cu atoms were latent in the reaction, and no byproducts were consequently formed. The apparent change in the properties of all Cu sites is actually a result of the dominant contribution of the Sb/Pd co-modified Cu atoms in Cu₉₂Sb₅Pd₃. Hence, our single-atom alloying strategy does not affect all Cu sites uniformly; rather, it creates selective active sites for the CO₂RR.

Fig. R1 | Schematic illustration of CO_2 -to- CO conversion on $\text{Cu}_{92}\text{Sb}_5\text{Pd}_3$. The light yellow Cu atoms adjacent to both the Sb and Pd atoms are regarded as true active sites.

Fig. 4 | **c**, CO_2RR to CO on $\text{Cu}_{92}\text{Sb}_5\text{Pd}_3$. **d**, Theoretical rates of CO production on $\text{Cu}_{92}\text{Sb}_5\text{Pd}_3$ compared with three control samples.

Supplementary Fig. 34 | **a**, CO_2RR to CO on Cu (211).

To confirm the above presumption, we have employed many elaborate experiments to explore the active site (i.e., *in situ* Raman, *in situ* ATR-SEIRAS, and *ex situ* CO-DRIFTS measurements), and conducted theoretical calculations to further verify this. As shown in **Figs. 3a** and **b** (copied below), noticeable Raman peaks at $\sim 2080\text{ cm}^{-1}$ attributed to CO^* at step sites of the Cu base emerged for four samples, demonstrating that Cu is the active site rather than a single-atom component. The low-frequency band at $\sim 2045\text{ cm}^{-1}$ corresponding to CO^* on terrace sites of the Cu base appeared only in three control samples, indicating greater coverage of CO^* on those samples. Moreover, during *in situ* Raman analysis, we also detected a peak at $\sim 360\text{ cm}^{-1}$ associated with Cu-CO stretching, further proving that the active sites are Cu atoms rather than single-atom additions. Then, we explored the adsorption behavior of the chemical intermediates at the active sites using *in situ* ATR-SEIRAS. Four bands related to surface-bonded CO^* at $2000\text{-}2100\text{ cm}^{-1}$ were attributed to Cu species [according to previous reports from Prof. Bingjun Xu and Prof. Wenjie Shen; *Nat. Catal.* **6**, 885-894 (2023); *Nat. Catal.* **2**, 334-341 (2019); *Nat. Catal.* **2**, 142-148 (2019); *Nat. Commun.* **13**, 2656 (2022)] appeared in the ATR-SEIRAS spectra (**Supplementary Fig. 26**, copied below), together with the main peaks at $\sim 2100\text{ cm}^{-1}$ attributed to CO_{ad} on Cu species in the CO-DRIFTS measurements (**Fig. 3c**, copied below), all of which proved that Cu sites serve as active sites for the CO_2RR rather than Sb or Pd single atoms.

Figs. 3a and b | *In situ* Raman spectra of four different catalysts at various potentials.

Supplementary Fig. 26 | In situ ATR-SEIRAS spectra of four samples during the CO₂RR.

Fig. 3c | CO-DRIFTS measurements of four different catalysts.

Regarding the concern about the effectiveness of small amounts of Sb and Pd dopants on the Cu matrix, we would like to mention that other Cu-based SAA catalysts with 3~5% single-atom dopants have also achieved high selectivity for CO₂ reduction to CO or formate [*Nat. Nanotechnol.* **16**, 1386-1393 (2021); *Nat. Commun.* **12**, 1449 (2021)]. Therefore, even though the Sb and Pd contents in the Cu₉₂Sb₅Pd₃ catalyst were only 5.0 and 3.0 at%, respectively, these amounts are sufficient to modify the copper substrate to provide optimal activity for CO₂-to-CO conversion. This phenomenon also aligns with the nature of single-atom catalysis, where even a sparse dopant distribution can exert a profound influence on catalytic sites [*J. Am. Chem. Soc.* **141**, 16635-16642 (2019); *ACS Energy Lett.* **6**, 2, 713-727 (2021); *Acc. Chem. Res.* **52**, 3, 656-664 (2019)].

Comment 2

FE of CO at -1 A cm⁻² is 85% in the main text, and therefore the claim of “~100% CO selectivity” in the abstract could be misleading.

Response

We thank the reviewer for this kind suggestion. To avoid any confusion, we have modified the statement to: “*This trimetallic single-atom alloy catalyst (Cu₉₂Sb₅Pd₃) achieves an impressive CO selectivity of ~100% at -402 mA cm⁻² and a high activity up to -1 A cm⁻² in a neutral electrolyte, surpassing most of the state-of-the-art noble metal catalysts.*”. We hope that these changes will improve the rigor of our work.

Comment 3

Although the performance of Cu₉₂Sb₅Pd₃ surpasses most of Au and Ag catalysts, noble metal Pd is used in this catalyst makes its.

Response

It seems that for several reasons, the respected reviewer did not fully state his/her comment. We infer that the reviewer’s comment concerns the role of the Pd component in the Cu₉₂Sb₅Pd₃ catalyst during the CO₂RR. We would like to clarify two points in response to this comment.

First, we explained in our response to *Comment 1* that single Pd atoms are not the real active sites of our Cu₉₂Sb₅Pd₃ catalyst. Instead, **the active sites are the Cu atoms that are modified by both Sb and Pd single-atom additions.** The Pd component does not directly

participate in the CO₂RR, but rather acts as an electron donor and a stabilizer for the Cu-Sb-Pd alloy structure. Therefore, the improved catalytic performance of Cu₉₂Sb₅Pd₃ is not due to the Pd sites on the Cu matrix but rather to the synergistic effect of the Sb and Pd dopants on the Cu surface. Considering this, the novelty and significance of our work lies in the design and synthesis of a Cu-based alloy catalyst that achieves ultrahigh activity, selectivity, and stability for the CO₂RR, **overcoming the limitations of pure Cu catalysts that often show poor selectivity and activity for a specific product** [*Energ. Environ. Sci.* **3**, 1311 (2010); *Nat. Catal.* **2**, 198 (2019)].

Second, we compared the selectivity of our Cu₉₂Sb₅Pd₃ catalyst with that of Pd-based catalysts for CO₂-to-CO conversion based on published literature. As shown in **Fig. R2**, our Cu₉₂Sb₅Pd₃ catalyst exhibited greater CO selectivity than most of the Pd-based catalysts over a wide potential range, indicating its exceptional catalytic performance. Moreover, we note that most Pd-based catalysts suffer from CO poisoning, which limits their activity and stability for the CO₂RR [*J. Am. Chem. Soc.* **141**, 16635–16642 (2019)]. In contrast, our Cu₉₂Sb₅Pd₃ catalyst demonstrated excellent durability for more than 528 hours without degradation. In addition, Pd has been reported to have very strong affinities for hydrogen species [*Nano Lett.* **22**, 11, 4576-4582 (2022)], resulting in a significant HER during the CO₂RR; however, in our case, FE_{H₂} can be suppressed to 7% at -1 A cm⁻². Thus, our Cu₉₂Sb₅Pd₃ catalyst is more suitable and competitive for the CO₂RR than Pd-based catalysts.

Fig. R2 | Comparison of the selectivities of Cu₉₂Sb₅Pd₃ and Pd-based catalysts in the CO₂RR. The catalyst references used were reproduced from: Pd NPs [*J. Am. Chem. Soc.* **137**, 4288-4291 (2015)], Pd-NC [*Adv. Funct. Mater.* **30**, 2000407 (2020)], Pd@Au [*J. Am. Chem. Soc.* **142**, 44, 18971-18980

(2020)], Pd₁Cu₁ [*Sep. Purif. Technol.* **320**, 124186 (2023)], 3.7 nm Pd [*Nano Res.* **10**(6), 2181–2191 (2017)] and Pd Octahedra [*Adv. Energy Mater.* **9**, 1802840 (2019)].

Comment 4

*It is well established that catalysts with weak *CO adsorption energy is favorable for CO₂ to CO in the experimental and computation literature. The authors devoted much of the characterization results (Fig.3) and discussion to re-prove this point, which doesn't yield much new insights.*

Response

Thank you for the comment. However, we believe that the reviewer misunderstood the purpose and significance of our characterization results in **Fig. 3**. We would like to explain and justify our choice of methods and analysis as follows:

As shown in **Figs. 3a-d**, we did not intend to re-prove the well-established relationship between the weak CO adsorption energy and high CO selectivity, but rather to gain a deeper understanding of the **CO₂-to-CO pathway** and the **different behaviours of the reactive intermediates** on the four as-synthesized SAA catalysts. We used *in situ* spectroscopic techniques to probe the active sites and intermediates of the CO₂RR and found that the active sites are the Cu atoms that are modified by the Sb and Pd single-atom dopants. This finding is novel and important for the design and optimization of SAA catalysts for the CO₂RR.

As shown in **Fig. 3e**, we further confirmed our hypothesis by direct SVBS evidence, which revealed a correlation between the downshifted *d*-band center and the weakened binding strength of CO intermediates. The SVBS experiment not only validates the literature but also demonstrates the successful **modification of the electronic state** of the Cu matrix by the single-atom alloying strategy. This strategy is the key innovation of our work and the main reason for the enhanced performance of our Cu₉₂Sb₅Pd₃ catalyst.

In addition, the adsorption of CO intermediates is not the only factor influencing the CO₂RR; moreover, the HER suppression and optimized CO onset potential are related to the electronic state of the Cu matrix. As shown in **Figs. 2d** and **e** (copied below), our Cu₉₂Sb₅Pd₃ catalyst has a **lower HER activity** and a **lower CO onset potential** than the other control catalysts, which indicates its superior selectivity and activity for the CO₂RR. These results are also supported by our experimental and theoretical studies in **Fig. 4** and **Fig. 5**, respectively.

Fig. 2d | CV investigations of the hydrogen desorption of the Cu and $\text{Cu}_{92}\text{Sb}_5\text{Pd}_3$ catalysts.

Fig. 2e | In situ DEMS measurements of four different catalysts in the CO_2RR .

Considering the above, we hope our response can fully convey our intention of combining multiple characterizations in **Fig. 3**.

Comment 5

*Pd possesses stronger *CO binding energy than Cu. Why the doping of Pd leads to the increase of desorption rates of COad in Fig.3d?*

Response

We appreciate the reviewer's comment. According to a previous study [*Phys. Chem. Chem. Phys.* **9**, 2216-2225 (2007)], the CO adsorption energy at the top sites of Cu-Pd alloy surfaces decreases with increasing concentrations of Pd, which contradicts expectations based on the stronger CO binding energy of Pd than of Cu. This phenomenon is explained by the effective compressive strain induced by the larger size of the Pd atoms, which weakens the CO-Pd bond. Therefore, the CO* binding energy of Pd is not a reliable indicator of CO adsorption on Cu-Pd alloys.

In addition, in our work, we used a Cu₉₂Sb₅Pd₃ catalyst for the CO₂ reduction reaction, in which **Pd did not act as a CO adsorption** site but rather modified the electronic structure of neighboring Cu atoms together with Sb single-atom dopants. The Sb and Pd components in the catalyst are electron acceptors that tune the catalytic activity of the Cu atoms adjacent to both the Sb and Pd single-atom additions. These **Cu atoms are the true active sites** of the catalyst, as we have demonstrated by various experimental (i.e., *in situ* Raman, *in situ* ATR-SEIRAS, and CO-DRIFTS) and theoretical methods. Moreover, according to the literature from Prof. Shyam Kattel, Prof. Bingjun Xu, and Prof. Jinguang Chen [*Nano Lett.* **22**, 11, 4576-4582 (2022)], when Pd serves as a catalytic site in the CO₂RR, it exhibits extremely strong affinity for hydrogen species (i.e., forming Pd hydride). If Pd atoms were the active sites for the CO₂RR in the Cu₉₂Sb₅Pd₃ catalyst, we would expect a significant HER as a side reaction, since Pd is known to be a highly active catalyst for the HER. However, our cyclic voltammetry investigations of the HER shown in **Fig. 2d** reveal that the Cu₉₂Sb₅Pd₃ catalyst has negligible HER activity, confirming that Pd is not involved in the CO₂RR.

Fig. 2d | CV investigations of the hydrogen desorption of the Cu and Cu₉₂Sb₅Pd₃ catalysts.

Furthermore, we attributed the main peak at $\sim 2100\text{ cm}^{-1}$ in **Fig. 3c** to CO_{ad} on Cu species based on literature reports [*Nat. Catal.* **2**, 334-341 (2019); *Nat. Catal.* **2**, 142-148 (2019); *Nat. Commun.* **10**, 5812 (2019)]. This is consistent with our *in situ* spectroscopy results, which show that CO adsorption occurs mainly on Cu sites in the four samples. Therefore, we concluded that the CO desorption on the Cu₉₂Sb₅Pd₃ catalyst measured by CO-DRIFTS occurred at the Cu sites rather than at the Pd sites. Moreover, the Pd content in the catalyst is only 3%, which is much lower than the Cu content (92%). This implies that CO adsorption is dominated by Cu sites, especially considering that only Cu serves as a catalytic site in the CO₂RR. Taken together, these findings indicate that the doping of Pd will not increase the desorption rate of CO_{ad}.

Fig. 3c | CO-DRIFTS measurements of four different catalysts.

Reviewer 2

In this work, the authors have synthesized a trimetallic single atom alloy catalyst (Cu₉₂Sb₅Pd₃) for the electrochemical reduction of CO₂ (CO₂RR). This Cu-based catalyst has shown an impressive CO selectivity (~100%) and a high Faradic efficiency (95%). The authors have investigated the roles of Sb and Pd single atoms via a series of spectroscopic techniques and theoretical simulations. It has been found that the presence of Sb and Pd atoms not only improved the stability of the alloy but also increased the selectivity of CO by decreasing the CO binding affinity on Cu atoms. While qualitative agreements between theoretical simulations and experimental measurements have been achieved, I believe that it would be more beneficial to the identification and understanding of the nature of the active center if more quantitative comparisons could be provided. In my opinion, this study is interesting and suitable for publication in Nature Communication after revision.

Response

We highly appreciate the reviewer's recognition of our study, as well as the constructive suggestions, which have greatly improved our studies. In this revised version of our manuscript, we have addressed all the questions raised by the reviewer, and supplemented more quantitative comparisons as well as spectroscopic analysis to resolve the reviewer's major concern, which has greatly improved the depth and rigor of our work.

Comment 1

A key concern is the spatial distribution of Sb and Pd single atoms, in particular, their preferences for the surface or the subsurface. Providing evidence or discussion on this aspect would enhance the clarity of the catalyst's structures.

Response

We appreciate the kind reviewer for this important suggestion. We completely understand the reviewer's concern over the distribution of single Pd/Sb atoms, and the STEM-EDS mapping of Cu, Sb and Pd in the Cu₉₂Sb₅Pd₃ catalyst are provided in **Fig. 1a** to demonstrate that two single-atom components are evenly dispersed in the bulk phase of the Cu matrix. To be more conspicuous, here, we re-illustrated the mapping images in **Fig. R3**, in which **no preference for surface or subsurface alloys was observed**. Considering that the scale bar is merely 5 nm, which is much smaller than the scale of the as-synthesized catalysts (10 to 20 nm, as shown by the TEM image in **Supplementary Fig. 4**, copied below), the occurrence of surface alloying or uneven distribution of Sb and Pd single atoms can be easily observed. If

Sb and Pd single atoms were preferentially located on the surface or subsurface of the catalyst, we would expect to see a contrast or a gradient in the mapping images, which is not the case. Since no such phenomenon was observed, the **Pd and Sb single-atom components were believed to exhibit uniform distributions in the bulk phase rather than being surface alloyed.**

Fig. R3 | STEM-EDS mapping of Cu, Sb and Pd in the $\text{Cu}_{92}\text{Sb}_5\text{Pd}_3$ catalyst.

Supplementary Fig. 4 | TEM images of the as-synthesized $\text{Cu}_{92}\text{Sb}_5\text{Pd}_3$ catalyst.

In addition, for our synthesis method, an exceedingly concentrated NaBH_4 solution was applied to reduce the metal salts. Part of the reason for using such a high concentration is that the violent reduction reaction yields a non-equilibrium condition during synthesis, leading to the instantaneous nucleation of the as-synthesized catalysts. Since equilibrium syntheses are generally thermodynamically controlled [*Nat. Synth.* (2023), <https://doi.org/10.1038/s44160-023-00387-3>], which might result in an uneven distribution of the elements. Our synthesis strategy potentially ensures the **uniform dispersion of Pd/Sb components throughout the whole bulk phase instead of just being confined to the surface.**

In conclusion, based on both the STEM-EDS mapping results and our synthesis strategy, it can be ensured that single Pd/Sb atoms are evenly distributed in the bulk phase of the catalyst rather than merely surface alloying on the cover, which addresses the reviewer's concern.

Comment 2

As compared to the other samples, the high CO-selectivity of the $\text{Cu}_{92}\text{Sb}_5\text{Pd}_3$ catalyst has been attributed to its weaker CO binding affinity, which strongly depends on the active site model where the CO is adsorbed. The structure of the active site, however, is solely determined by the comparison of calculated CO adsorption energies on three different models with other samples (Cu, $\text{Cu}_{97}\text{Pd}_3$, and $\text{Cu}_{95}\text{Sb}_5$). Is there any direct correspondence between the theory and experiment for the structure of the active site (e.g. from the spectroscopic evidence)?

Response

Thank you for your comment. We not only compared the calculated CO adsorption energies of three different models with those of other control samples but also validated our models by comparing them with the experimental results. As shown in **Fig. 4a** (copied below), the calculated CO adsorption energy on Cu₉₂Sb₅Pd₃ (211) is much greater than that on the Cu₉₂Sb₅Pd₃ (211)-1 and Cu₉₂Sb₅Pd₃ (211)-2 models and is even greater than that on the three control samples, i.e., Cu, Cu₉₇Pd₃ and Cu₉₅Sb₅. However, this trend contradicts the experimental CO selectivities of the four catalysts, which shows that Cu₉₂Sb₅Pd₃ has the best catalytic performance in the CO₂RR. Therefore, we excluded Cu₉₂Sb₅Pd₃ (211) as a possible model for the active surface of the Cu₉₂Sb₅Pd₃ catalyst, and focused on the other two models that agreed better with the experimental data. In addition, the CO-DRIFTS data in **Fig. 3d** (copied below) also prove that the desorption rates of CO_{ad} among the four samples rank as follows: $r_{\text{CO}}(\text{Cu}_{92}\text{Sb}_5\text{Pd}_3) > r_{\text{CO}}(\text{Cu}_{95}\text{Sb}_5) > r_{\text{CO}}(\text{Cu}_{97}\text{Pd}_3) > r_{\text{CO}}(\text{Cu})$. The CO adsorption energy on Cu₉₂Sb₅Pd₃ is supposed to be lower than that on the other control samples. As such, Cu₉₂Sb₅Pd₃ (211) was unlikely to be the correct model. In addition, to determine the major active structure of the Cu₉₂Sb₅Pd₃ catalyst, we calculated the theoretical rates of CO production from Cu₉₂Sb₅Pd₃ (211)-1 and Cu₉₂Sb₅Pd₃ (211)-2, as shown in **Fig. 4d** (copied below). Based on previous experimental results, the CO selectivity of Cu₉₂Sb₅Pd₃ is much greater than that of Cu₉₅Sb₅; therefore, their theoretical rates of CO production should exhibit distinct differences. As Cu₉₂Sb₅Pd₃ (211)-1 only slightly improved upon Cu₉₅Sb₅ in CO production, Cu₉₂Sb₅Pd₃ (211)-2 was believed to be the major active structure. The above theoretical results indicate that our models are reasonable and reliable for describing the CO₂RR on the Cu₉₂Sb₅Pd₃ catalyst.

We also used spectroscopic techniques to confirm the structure of the active site on the Cu₉₂Sb₅Pd₃ catalyst. As shown in **Fig. 3a** and **Supplementary Fig. 26**, we performed *in situ* Raman and *in situ* ATR-SEIRAS measurements to probe the CO₂ adsorption and reduction intermediates on the four as-synthesized SAA catalysts. In the Raman spectra (**Fig. 3a**), the band at ~2080 cm⁻¹ attributed to CO* on the step sites of the Cu base was observed for all the samples, demonstrating that Cu is the active site rather than a single-atom component. On the other hand, four bands related to surface-bonded CO* at 2000-2100 cm⁻¹ attributed to Cu species [*Nat. Catal.* **2**, 334-341 (2019); *Nat. Catal.* **2**, 142-148 (2019); *Nat. Commun.* **10**, 5812 (2019)] appeared in the ATR-SEIRAS spectra (**Supplementary Fig. 26**), together with the

main peak at $\sim 2100\text{ cm}^{-1}$ attributed to CO_{ad} on Cu species in the CO-DRIFTS measurements (**Fig. 3c**). These findings confirmed that copper sites serve as adsorption sites for CO in the four samples, reconfirming the accuracy of our theoretical model. Moreover, the intensity of the CO_{ad} peak for the $\text{Cu}_{92}\text{Sb}_5\text{Pd}_3$ catalyst in **Supplementary Fig. 26** is much lower than that for the other SAA catalysts, indicating a weaker CO binding affinity, which is also in agreement with our theoretical calculations.

Therefore, we have provided direct correspondence between the theoretical and experimental results for the structure of the active site on our $\text{Cu}_{92}\text{Sb}_5\text{Pd}_3$ catalyst. We have shown that the active site is the Cu atom that is modified by both Sb and Pd single-atom additions, and that this Cu atom has a weaker CO binding affinity than the other Cu-based catalysts, which leads to greater CO selectivity in the CO_2RR . Specifically, $\text{Cu}_{92}\text{Sb}_5\text{Pd}_3$ (211)-2 is supposed to be the major active structure, while $\text{Cu}_{92}\text{Sb}_5\text{Pd}_3$ (211)-1 tends to be suboptimal.

Fig. 4a | The calculated adsorption energy and structures of CO^* on three models.

Fig. 3d | CO-DRIFTS measurements of four different catalysts.

Fig. 4d | Theoretical rates of CO production on Cu₉₂Sb₅Pd₃ (211)-1 and Cu₉₂Sb₅Pd₃ (211)-2.

Fig. 3a | *In situ* Raman spectra of $\text{Cu}_{92}\text{Sb}_5\text{Pd}_3$ at different potentials.

Supplementary Fig. 26 | *In situ* ATR-SEIRAS spectra of four samples during the CO_2RR .

Comment 3

The authors have investigated three types of trimetallic active structures by DFT calculations. According to the CO binding affinity and the activity, the authors suggested the most active model $\text{Cu}_{92}\text{Sb}_5\text{Pd}_3(211)\text{-2}$ as the active site, while the $\text{Cu}_{92}\text{Sb}_5\text{Pd}_3(211)$ model is excluded because of its stronger CO binding affinity as compared to Cu. Does the existence of other types of trimetallic active structures (e.g. $\text{Cu}_{92}\text{Sb}_5\text{Pd}_3(211)$) influence the reaction rate of side reactions and the selectivity? The authors should provide more explanations on why Cu(211) was employed in the modeling.

Response

We thank the reviewer for raising this issue. Based on the reviewer's suggestions, we have made additional calculations, where the CO₂RR and HER processes on the three models of Cu₉₂Sb₅Pd₃ were considered. The barriers and reaction free energies involved are summarized in **Table R1**. The reaction rates of the main and side reactions were calculated at -0.93 V vs. RHE, as listed in **Table R2**. The side reactions are mainly formic acid production and the HER. Although the side reaction rates of Cu₉₂Sb₅Pd₃ (211) (TOF_{HCOOH}=1.63×10² s⁻¹; TOF_{H₂}=8.46×10³ s⁻¹) are greater than those of the main reaction (TOF_{CO}=1.61×10⁰ s⁻¹), they are much lower than the main reaction rates of Cu₉₂Sb₅Pd₃ (211)-1 (TOF_{CO}=3.62×10⁵ s⁻¹) and Cu₉₂Sb₅Pd₃ (211)-2 (TOF_{CO}=7.97×10⁵ s⁻¹). In addition, the surface energies of Cu₉₂Sb₅Pd₃ (211) (0.19) and Cu₉₂Sb₅Pd₃ (211)-1 (0.19) are greater than that of Cu₉₂Sb₅Pd₃ (211)-2 (0.18), indicating that the probability of existence of Cu₉₂Sb₅Pd₃ (211)-2 is the highest. In other words, the probability of the existence of Cu₉₂Sb₅Pd₃ (211) is relatively low. In summary, Cu₉₂Sb₅Pd₃ (211)-2 is the main active site for generating CO, while the presence of the other two structures has a relatively small impact on the selectivity for CO.

Table R1 | Calculated barriers (G_a) and reaction energies (ΔG) of the considered elementary steps for the CO₂RR and HER on the Cu₉₂Sb₅Pd₃ (211), Cu₉₂Sb₅Pd₃ (211)-1 and Cu₉₂Sb₅Pd₃ (211)-2 surfaces at -0.93 V vs. RHE.

Reaction	Cu ₉₂ Sb ₅ Pd ₃ (211)		Cu ₉₂ Sb ₅ Pd ₃ (211)-1		Cu ₉₂ Sb ₅ Pd ₃ (211)-2	
	G_a	ΔG	G_a	ΔG	G_a	ΔG
CO ₂ + (H ⁺ +e ⁻) + * → COOH*	0.44	0.31	0.43	0.10	0.41	0.11
COOH*+(H ⁺ +e ⁻)→CO*+H ₂ O	0.00	-1.98	0.12	-1.38	0.11	-1.37
CO* → CO + *	0.48	0.48	0.09	0.09	0.07	0.07
CO ₂ + (H ⁺ +e ⁻) + * → HCOO*	0.63	-0.45	0.54	-0.66	0.62	-0.68
HCOO*+(H ⁺ +e ⁻)→HCOOH+*	0.00	-0.61	0.00	-0.40	0.00	-0.39
(H ⁺ +e ⁻) + * → H*	0.51	-0.87	0.75	-0.86	0.63	-0.83
H* + (H ⁺ +e ⁻) → H ₂ + *	0.51	-0.99	0.25	-1.00	0.24	-1.03

Table R2 | TOFs of different products for the CO₂RR and HER on the Cu₉₂Sb₅Pd₃ (211), Cu₉₂Sb₅Pd₃ (211)-1 and Cu₉₂Sb₅Pd₃ (211)-2 surfaces at -0.93 V vs. RHE.

	CO (s ⁻¹)	C ₂₊ (s ⁻¹)	HCOOH (s ⁻¹)	H ₂ (s ⁻¹)
Cu ₉₂ Sb ₅ Pd ₃ (211)	1.61×10 ⁰	8.91×10 ⁻⁴	1.63×10 ²	8.46×10 ³
Cu ₉₂ Sb ₅ Pd ₃ (211)-1	3.62×10 ⁵	3.79×10 ⁻⁴	5.30×10 ³	1.57×10 ⁰

Cu ₉₂ Sb ₃ Pd ₃ (211)-2	7.97×10 ⁵	9.22×10 ⁻⁹	2.40×10 ²	1.63×10 ²
--	----------------------	-----------------------	----------------------	----------------------

Regarding the reason for employing Cu (211) in the modelling, previous work has compared the effects of Cu (111), Cu (100) and Cu (211) surfaces on the activation of the electroreduction of CO₂ [*Surf. Sci.* **605**, 1354-1359 (2011)]. The formation of adsorbed carboxyl (COOH*) is the key elementary step involved in the formation of CO. They found that COOH* is bound to the (211) step much more strongly than to the (100) and (111) surfaces. This thermodynamic analysis indicates that a lower potential is required to convert CO₂ to adsorbed CO on the (211) surface. This leads to the prediction that the (211) facet is the most active surface among the three surfaces for producing CO. For copper-based single-atom alloy catalysts, their simulation performance on the step surface is in good agreement with the experimental results [*Nat. Commun.* **14**, 340 (2023); *Nat. Nanotechnol.* **16**, 1386-1393 (2021)]. Thus, the Cu (211) surface model was finally chosen. As suggested by the reviewer, the above explanations have been supplemented into the article.

Comment 4

I recommend the authors to provide more discussions on the individual roles of Sb and Pd in modifying catalytic performance and the synergistic cooperation between them.

Response

We thank the reviewer for his/her thoughtful suggestion. For the role of single-atom components in modifying catalytic performance, we interpret the individual and synergistic effects of Sb and Pd components separately.

First, individually, either the Sb or Pd component is capable of suppressing C-C coupling on the Cu matrix by modifying the electronic structure of Cu to a proper state. As demonstrated by the SVBS measurements (**Fig. 3e**, copied below), the *d*-band centers both shifted downwards after Sb/Pd single atoms were separately added to the Cu base. Specifically, the 5% Sb single-atom component could modify the Cu base to a greater degree than the 3% Pd addition. However, the catalytic performances of both bimetallic counterparts have an upper limit. Neither could achieve exclusive CO selectivity under large current densities, as shown in **Fig. 2a** (copied below).

Fig. 3e | SVBS measurements of four as-synthesized catalysts.

Fig. 2a | FEs of CO₂RR products at different current densities for Cu₉₂Sb₅Pd₃, Cu₉₅Sb₅ and Cu₉₇Pd₃.

In terms of the synergistic effect, once Sb/Pd single atoms were co-introduced into the alloy system, a collaborative effect became noticeable. The *d*-band center of Cu₉₂Sb₅Pd₃ reached the lowest level in **Fig. 3e**, and its ability to promote CO₂RR performance has also been confirmed experimentally and theoretically (**Fig. 2a**). Although one single-atom component adjusts the electronic structure of the Cu matrix to some extent, further addition of the other component fine-tunes its state. An overall result turned out to be an inhibition on side reactions (*i.e.*, CO₂-to-C₂/formate generation or HER), and that CO formation was promoted by improving CO₂ activation and weakening the binding strength of CO* intermediates. In addition, a synergistic effect on the catalytic performance of Cu₉₂Sb₅Pd₃ was also observed because of its extremely elevated stability for up to 528 h compared to that of its bimetallic counterpart Cu₉₅Sb₅, as the latter segregated after electrolysis due to structural instability. Our theoretical simulations (**Supplementary Fig. 25**, copied below) revealed the lowest surface

energy of 0.18 eV per atom for Cu₉₂Sb₅Pd₃, reconfirming the improved stability of the Cu₉₂Sb₅Pd₃ SAA catalyst by co-doping Sb and Pd on a Cu base.

Supplementary Fig. 25 / Surface energies of Cu, Cu₉₇Pd₃, Cu₉₅Sb₅, and Cu₉₂Sb₅Pd₃.

Comment 5

While the TOFs of different samples/models have been calculated via theoretical simulations, they cannot be compared to experimental measurements directly. It would be more convincing if the authors calculate and compare the TOFs of side reactions with the TOF of the major reaction, such that the selectivity can be obtained and compared with experimental measurements.

Response

We thank the reviewer for this useful advice, based on which we supplemented additional calculations, where the CO₂RR and HER processes on the Cu₉₇Pd₃ (211), Cu₉₅Sb₅ (211) and Cu₉₂Sb₅Pd₃ (211)-2 surfaces were considered. The barriers and reaction free energies of the main and side reactions are summarized in **Table R3**. The TOFs of the different products at -0.93 V vs. RHE are listed in **Table R4**. The FE was described by following equation:

$$\text{FE}/\% = \frac{n(i)\text{TOF}(i)}{\sum n(i)\text{TOF}(i)} \times 100$$

where $n(i)$ represents the electron transfer number and $\text{TOF}(i)$ is the turnover frequency obtained by microkinetic simulation for product i . As shown in **Supplementary Fig. 35**, the calculated FE_{CO} (blue bar) follows the order of Cu₉₇Pd₃ (211) < Cu₉₅Sb₅ (211) < Cu₉₂Sb₅Pd₃ (211)-2, which is comparable to the experimental results (red bar) for all three catalysts.

Table R3 | Calculated barriers (G_a) and reaction energies (ΔG) of the considered elementary steps for the CO₂RR and HER on the Cu₉₇Pd₃ (211), Cu₉₅Sb₅ (211) and Cu₉₂Sb₅Pd₃ (211)-2 surfaces at -0.93 V vs. RHE.

Reaction	Cu ₉₇ Pd ₃ (211)		Cu ₉₅ Sb ₅ (211)		Cu ₉₂ Sb ₅ Pd ₃ (211)-2	
	G_a	ΔG	G_a	ΔG	G_a	ΔG
CO ₂ + (H ⁺ +e ⁻) + * → COOH*	0.44	0.03	0.43	0.05	0.41	0.11
COOH* + (H ⁺ +e ⁻) → CO* + H ₂ O	0.12	-1.40	0.18	-1.36	0.11	-1.37
CO* → CO + *	0.18	0.18	0.11	0.11	0.07	0.07
CO ₂ + (H ⁺ +e ⁻) + * → HCOO*	0.51	-0.81	0.59	-0.73	0.62	-0.68
HCOO*+(H ⁺ +e ⁻) → HCOOH + *	0.00	-0.25	0.00	-0.34	0.00	-0.39
(H ⁺ +e ⁻) + * → H*	0.59	-0.83	0.73	-0.82	0.63	-0.83
H* + (H ⁺ +e ⁻) → H ₂ + *	0.13	-1.03	0.23	-1.04	0.24	-1.03

Table R4 | TOFs of different products for the CO₂RR and HER on Cu₉₇Pd₃ (211), Cu₉₅Sb₅ (211) and Cu₉₂Sb₅Pd₃ (211)-2 at -0.93 V vs. RHE.

	CO (s ⁻¹)	C ₂₊ (s ⁻¹)	HCOOH (s ⁻¹)	H ₂ (s ⁻¹)
Cu₉₇Pd₃ (211)	1.23×10 ⁵	5.25×10 ³	1.15×10 ⁴	7.66×10 ²
Cu₉₅Sb₅ (211)	3.49×10 ⁵	2.74×10 ⁰	7.66×10 ²	3.41×10 ⁰
Cu₉₂Sb₅Pd₃ (211)-2	7.97×10 ⁵	9.22×10 ⁻⁹	2.40×10 ²	1.63×10 ²

Supplementary Fig. 35 | Comparison between the calculated FE_{CO} from microkinetic simulations on Cu₉₇Pd₃ (211), Cu₉₅Sb₅ (211) and Cu₉₂Sb₅Pd₃ (211)-2 and the experimental FE_{CO} on Cu₉₇Pd₃, Cu₉₅Sb₅ and Cu₉₂Sb₅Pd₃.

Comment 6

In this manuscript (line 83), the mix entropy was emphasized. On the other hand, providing more illustrations regarding the contributions of enthalpy, particularly interactions among metals are desirable, as they often play a significant role in determining free energy changes.

Response

Thank you for your constructive comment. We agree that the contribution of the enthalpy (ΔH) is important for determining the free energy changes in the alloy system. As reported in previous literature [*npj. Comput. Mater.* **4**, 47 (2018)], well-mixed multi-component alloys have relatively shallow negative ΔH values among the constituting elements, which prevents phase separation. Moreover, due to the high entropy of these alloys, the entropy contribution ($T\Delta S$) can be significant. Therefore, in our case, entropy plays a more dominant role in phase stability than does ΔH .

We also provide additional details on the mixing entropy (ΔS_{mix}) of different catalysts in this work. For comparison, we calculated ΔS_{mix} using the following equation [*ACS Catal.* **10**, 11280-11306 (2020)]:

$$\Delta S_{\text{mix}} = -R \cdot \sum X_i \cdot \ln(X_i)$$

where R is the molar gas constant, and X_i represents the mole ratio of the single-atom components in the alloy system. Given the above, the as-calculated ΔS_{mix} values of the three samples are $0.1347 R$ ($\text{Cu}_{97}\text{Pd}_3$), $0.1985 R$ ($\text{Cu}_{95}\text{Sb}_5$), and $0.3317 R$ ($\text{Cu}_{92}\text{Sb}_5\text{Pd}_3$). The mixing entropy increases as the number or ratio of single-atom components increases.

Using the formula for free energy ($\Delta G = \Delta H - T\Delta S$), we can infer that when ΔH is shallow and similar among the three SAA catalysts, the ΔG of the alloy system decreases as ΔS_{mix} increases with additional single-atom additions.

As suggested by the reviewer, we have added the above discussion to the **Supplementary Note** in the revised manuscript.

Comment 7

The authors should provide more details of “charge extrapolation” method within the capacitor model. Also providing the atomic coordinates of DFT calculations is encouraged.

Response

Thank you for your kind suggestion. The electrochemical barriers (G_a) were calculated on the basis of the “charge-extrapolation” method [*J. Phys. Chem. Lett.* **7**, 1686-1690 (2016)]

within the capacitor model. The amount of electron transfer (Δq) from the water layer to the electrode is linearly correlated with the relative work function (Φ) at the initial state (IS), transition state (TS), and final state (FS), as shown in **Supplementary Figs. 30-33**. According to the capacitor model, the energy change between two states at a constant work function can be calculated as follows:

$$E_2(\Phi_1) - E_1(\Phi_1) = E_2(\Phi_2) - E_1(\Phi_1) + \frac{(q_2 - q_1)(\Phi_2 - \Phi_1)}{2}$$

$$E_2(\Phi_2) - E_1(\Phi_2) = E_2(\Phi_2) - E_1(\Phi_1) - \frac{(q_2 - q_1)(\Phi_2 - \Phi_1)}{2}$$

where $E_1(\Phi_1)$ and $E_2(\Phi_2)$ correspond to the energies of states 1 and 2, respectively. Φ and q refer to the work function and interfacial charge transfer, respectively.

Setting $\Delta E(\Phi) = E_2(\Phi) - E_1(\Phi)$ at a given work function Φ and $\Delta q = q_2 - q_1$, the following equation can be derived:

$$\Delta E(\Phi_2) - \Delta E(\Phi_1) = -\Delta q(\Phi_2 - \Phi_1)$$

where $\Delta E(\Phi_1)$ and $\Delta E(\Phi_2)$ are the barriers at Φ_1 and Φ_2 , respectively. The work function Φ can be related to the absolute potential (U_{SHE}) by $U_{\text{SHE}} = \frac{\Phi - \Phi_{\text{SHE}}}{e}$, where Φ_{SHE} has been determined experimentally to be ~ 4.4 eV. Therefore, the potential-dependent barrier can be calculated by this method.

Supplementary Fig. 30 | Calculated charge transfer (Δq) and Φ on electrochemical interface at the initial state (IS), transition state (TS), and final state (FS) for COOH^* formation (a), HCOO^* formation (b), Volmer (c) and Heyrovsky (d) steps over $\text{Cu}_{97}\text{Pd}_3$ (211).

Supplementary Fig. 31 | Calculated charge transfer (Δq) and Φ on electrochemical interface at the initial state (IS), transition state (TS), and final state (FS) for COOH^* formation (a), HCOO^* formation (b), Volmer (c) and Heyrovsky (d) steps over $\text{Cu}_{95}\text{Sb}_5$ (211).

Supplementary Fig. 32 | Calculated charge transfer (Δq) and Φ on electrochemical interface at the initial state (IS), transition state (TS), and final state (FS) for COOH^* formation (a), HCOO^* formation (b), Volmer (c) and Heyrovsky (d) steps over $\text{Cu}_{92}\text{Sb}_5\text{Pd}_3$ (211)-2.

Supplementary Fig. 33 | Calculated charge transfer (Δq) and Φ on electrochemical interface at the initial state (IS), transition state (TS), and final state (FS) for COOH^* formation (a), HCOO^* formation (b), Volmer (c) and Heyrovsky (d) steps over $\text{Cu}_{92}\text{Sb}_5\text{Pd}_3$ (211)-1.

The atomic coordinates of the Cu (211), $\text{Cu}_{97}\text{Pd}_3$ (211), $\text{Cu}_{95}\text{Sb}_5$ (211), $\text{Cu}_{92}\text{Sb}_5\text{Pd}_3$ (211), $\text{Cu}_{92}\text{Sb}_5\text{Pd}_3$ (211)-1 and $\text{Cu}_{92}\text{Sb}_5\text{Pd}_3$ (211)-2 surfaces with adsorbed CO are provided as supporting materials.

Reviewer 3

This work presents an intriguing design principle for synthesizing a trimetallic catalyst via single-atom alloying strategy to achieve CO₂-to-CO conversion with excellent selectivity, activity, and stability. In their previous work (Nat. Commun. 14, 340, 2023), the group presented a bimetallic Sb-alloyed copper catalyst for CO production. But its production rate and stability are still inferior to those of noble metals like Au and Ag. As a follow-up, in this current work, the incorporation of mixed entropy concepts into the trimetallic Cu-Sb-Pd system results in a holistic enhancement and overall upgrade in CO₂ conversion, almost on par with Ag/Au. The synergy of Pd/Sb single atoms significantly improves the selectivity and activity of the catalyst by adjusting the d-band center of Cu to an optimal state. Additionally, this approach reinforces the resistance of catalyst to deactivation, as the optimized surface energy elevates the migration energy of the sites. In this regard, this work marks a noteworthy achievement in both catalytical performance and mechanism exploration. This paper is well written and provides valuable insights for the community, which is a truly breakthrough. I recommend its acceptance after addressing the following issues.

Response

We sincerely appreciate the reviewer's high evaluation of our study, as well as the important suggestions that have greatly improved our work. As suggested by the reviewer, we have implemented changes and refinements in the revised version. In addition, we thank the reviewer for pointing out and commenting on our previously published work, which clarified the significance and improvements of this work.

Comment 1

The authors showed that the control samples Cu₉₂Sb₈ and Cu₉₂Pd₈ have inferior selectivity toward CO compared to Cu₉₂Sb₅Pd₃ in Supplementary Fig.18. Given that the additional component contents are all equal to 8%, it would be helpful if the authors delve into the underlying reason for this observation. In particular, why does increasing the Pd content in Cu₉₂Pd₈ not suppress the C-C coupling on the Cu matrix? What is the role of the single-atom alloying strategy in tuning the selectivity of the catalyst?

Response

We appreciate the reviewer for this good point. To determine the intrinsic reason for the inferior catalytic performances of Cu₉₂Sb₈ and Cu₉₂Pd₈ in the CO₂RR, we conducted HAADF-STEM combined with EDS investigations on these control samples.

For the Cu₉₂Sb₈ catalyst, post-reaction characterization confirmed that the SAA structure was not maintained after electrolysis, as shown in **Fig. R5**. The segregation of Sb was

noticeable in the STEM-EDS image. This phenomenon also explains the enhancement in formate generation at large current densities, since the unmodified Sb component is known for its ability to convert CO₂ into formate [*Energy Environ. Sci.* **13**, 2856 (2020)]. As such, a higher content of Sb in the Cu₉₂Sb₈ catalyst might lead to a preference for the formate product.

Fig. R5 | Post-reaction characterization of the Cu₉₂Sb₈ catalyst.

For Cu₉₂Pd₈, Pd easily agglomerated into clusters after synthesis (**Fig. R6**). Even though an 8% content of the Pd component was introduced into Cu₉₂Pd₈, only a minority of it functioned as a single-atom additive to modulate the Cu matrix. Therefore, the C-C coupling on the Cu matrix became prominent.

Fig. R6 | Post-reaction characterization of the $\text{Cu}_{92}\text{Pd}_8$ catalyst.

To determine the role of a single-atom alloying strategy in tuning catalyst selectivity, we separately interpret the individual and synergistic effects of Sb and Pd components on tuning the selectivity of the Cu matrix. First, individually, either the Sb or Pd component is capable of suppressing C-C coupling on the Cu matrix by modifying the electronic structure of Cu to a proper state. As demonstrated by the SVBS measurements (**Fig. 3e**, copied below), the *d*-band centers both shifted downwards after Sb/Pd single atoms were added to the Cu base. However, the catalytic performances of these bimetallic counterparts have an upper limit. In terms of the synergistic effect, once they were co-introduced into the catalyst system, a collaborative effect became noticeable. The *d*-band center of $\text{Cu}_{92}\text{Sb}_5\text{Pd}_3$ reached its lowest level, and its ability to promote the CO_2RR performance has been confirmed experimentally and theoretically. Although the Sb single-atom component adjusts the electronic structure of the Cu matrix to some extent, further addition of the Pd constituent finely tunes its state. An overall result turned out to be an inhibition on side reactions (*i.e.*, $\text{CO}_2\text{-to-C}_2\text{/formate}$

generation or HER), and that CO formation was promoted by improving CO₂ activation and weakening the binding strength of CO* intermediates.

Fig. 3e | SVBS measurements of four as-synthesized catalysts.

Comment 2

The authors use “mixing entropy” to measure the free energy of the system, the terminology of which resembles the entropy-based definition of high-entropy alloys (ACS Catal. 2020, 10, 11280-11306). Is there any connection between the Cu₉₂Sb₅Pd₃ catalyst and HEA? Or can it be categorized as HEA or MEA (medium-entropy alloy)?

Response

We appreciate the reviewer’s insightful comment. We also considered the possible connection between HEA/MEA and our trimetallic Cu₉₂Sb₅Pd₃ catalyst at the beginning of our work. However, after reviewing the literature and performing careful calculations, we concluded that the Cu₉₂Sb₅Pd₃ catalyst is not an MEA or HEA. We explain the reasons as follows.

According to the reference cited by the reviewer, the mixing entropy (ΔS_{mix}) of the SAA catalysts in this work can be calculated by the following equation:

$$\Delta S_{\text{mix}} = - R \cdot \sum X_i \cdot \ln(X_i)$$

where R is the molar gas constant, and X_i represents the mole ratio of the single-atom components in the alloy system. Given the above, the as-calculated ΔS_{mix} values of the three samples are $0.1347 R$ (Cu₉₇Pd₃), $0.1985 R$ (Cu₉₅Sb₅), and $0.3317 R$ (Cu₉₂Sb₅Pd₃). The mixing entropy clearly increases as the number or ratio of single-atom components increases. According to the literature {Zhang Y. Amorphous and high entropy alloys [M]. Beijing:

Science Press, 2010: 68-71}, alloy materials can be classified into three categories according to the value of ΔS_{mix} , *i.e.*, high entropy alloy (HEA, $\Delta S_{\text{mix}} > 1.61 R$), medium entropy alloy (MEA, $0.69 R \leq \Delta S_{\text{mix}} \leq 1.61 R$) and low-entropy alloy (LEA, $\Delta S_{\text{mix}} < 0.69 R$). Since the ΔS_{mix} values of all three samples in our work fall into the LEA range, the $\text{Cu}_{92}\text{Sb}_5\text{Pd}_3$ catalyst cannot be categorized as MEA or HEA. This is reasonable, considering that the single-atom components in the trimetallic catalyst only account for a small fraction of the total.

Comment 3

Although this work establishes the outstanding catalytic performance of the $\text{Cu}_{92}\text{Sb}_5\text{Pd}_3$ catalyst in CO_2 reduction, schematic illustrations of the cell configurations used in the experiments are notably absent. To enhance the clarity of the methodology, I recommend providing additional details on the cell structures employed.

Response

We thank the reviewer for this kind advice. As suggested, we have added the cell configurations used in this work to the **Supporting Information (Supplementary Figs. 38 and 39, copied below)**. We believe such illustrations would better display our work.

Supplementary Fig. 38 | Schematic illustration of the flow-cell configuration.

Supplementary Fig. 39 | Schematic illustration of the MEA used in the stability test.

Comment 4

Considering the potential industrial applications of the $\text{Cu}_{92}\text{Sb}_5\text{Pd}_3$ catalyst, I recommend calculating cathodic energy efficiencies in a flow cell under applied potentials to further evaluate its viability for practical use.

Response

We thank the reviewer for this important suggestion. As suggested, we calculated the cathodic energy efficiency (EE) in **Fig. R7**. The following are the details of the calculations.

Fig. R7 | The calculated EE for CO₂-to-CO conversion on the Cu₉₂Sb₅Pd₃ catalyst.

Based on the above formula, the EE of the CO product can be calculated by:

$$EE (\%) = (1.23 - E_{\text{CO}}) \times FE_{\text{CO}} / (1.23 - E)$$

where E is the cathodic potential vs. RHE, and E_{CO} is the thermodynamic potential (-0.11 V vs. RHE) of the CO₂RR to CO. As such, the calculation results of the EE for CO₂-to-CO conversion on the Cu₉₂Sb₅Pd₃ catalyst under applied cathodic potentials are presented below. At -0.93 V vs. RHE, the EE reached 62% for the Cu₉₂Sb₅Pd₃ catalyst, with the highest FE_{CO} of ~100%.

Comment 5

In Supplementary Fig.26, the scale bars among four catalysts vary from each other. Although they don't impact the analysis and conclusion of the ATR-SEIRAS results, it would be preferred if the authors elucidate the cause for the difference.

Response

We appreciate the reviewer for his/her careful observation. The scale bars among the four catalysts in **Supplementary Fig. 26** vary from each other because the ATR-SEIRAS spectra were recorded at different magnifications to capture the best signal-to-noise ratio for each catalyst. The magnification factor was proportional to the surface roughness of the catalyst, which was affected by the composition and morphology of the single-atom catalysts. Therefore, the scale bars reflect the different surface characteristics of the four catalysts.

Comment 6

As the electrode preparation process might result in partial oxidation of the catalyst, clear descriptions of this process should be given in Methods.

Response

We thank the reviewer for raising this point. To clarify, we have updated the related statement in the **Methods** section as follows: “*To prepare the cathode electrode, precursor ink (12 mg of catalyst mixed with 24 μ L of 5% Nafion 117 solution dissolved in 2 mL of IPA) was spray-coated onto a gas diffusion layer (YLS-30T) with a mass loading of $\sim 1 \text{ mg cm}^{-2}$ using an air brush and eventually dried in air on a hotplate at 60 $^{\circ}\text{C}$.”. As such, we believe that the detailed description of the preparation process better explains the unavoidable oxidation of the catalyst.*

REVIEWER COMMENTS

Reviewer #1 (Remarks to the Author):

The reviewer appreciates the substantial effort in addressing the comments raised, but maintains the overall original assessment of the manuscript. Recent literature are full of examples of CO selective catalysts in the CO₂RR, including bulk metals, supported metal nanoparticles and single atom catalysts, so a Pd-containing catalyst (albeit with a low loading) hardly advances the field. On the mechanistic front, the authors' argument that this work provides novel mechanistic insights because they employed a number of characterization and computational methods does not hold water. With all the characterization and computational efforts, the proposed mechanism has been discussed repeatedly in the literature, i.e., introducing a second/third element changes the electronic structure and weakens the CO binding energy. The reviewer recognize the amount of effort dedicated to this work, however, is not optimistic about its impact on the relevant field. More detailed comments:

1) Regarding the response to Comment 1, the authors stated "The co-introduction of two single-atom components collaboratively alters the electronic state of neighbouring copper atoms, making them more favourable for CO formation and less prone to the HER", which is a reasonable hypothesis. The reviewer is curious that for many (or the majority of) Cu sites that are not close to Pd or Sb, how come they do not behave as typical metallic Cu sites, i.e., producing a wide range of C₂+ products?

2) Fig3 and Fig. S26 show inconsistent results of infrared and Raman results, i.e., CO band is absent on CuSbPd in Raman but present in infrared.

Reviewer #3 (Remarks to the Author):

The resolution of all the issues I have raised will undoubtedly serve as a valuable source of inspiration for the researchers involved. Therefore, I strongly recommend accepting it for publication.

Chuan Xia Mater. & Energy School
Professor of Chemistry UESTC, China
Phone: +86 028-61838256 2006 Xiyuan Ave,
Email: chuan.xia@uestc.edu.cn Chengdu, Sichuan 611731

Manuscript Number: **NCOMMS-23-57392**

Title: **“Turning copper into an efficient and stable CO evolution catalyst beyond noble metals”**

Authors: Jing Xue, Xue Dong, Chunxiao Liu, Jiawei Li, Yizhou Dai, Weiqing Xue, Laihao Luo, Yuan Ji, Xiao Zhang, Xu Li, Qiu Jiang, Tingting Zheng, Jianping Xiao, Chuan Xia

Corresponding authors: Jianping Xiao, Chuan Xia

Response to reviewers' comments:

We thank the editor and reviewers for their time and insightful comments on our work. We have now provided additional analyses to fully address **Reviewer 1**'s concerns. Below, we address all the remaining issues raised by the reviewer one by one in the following pages.

Reviewer 1

The reviewer appreciates the substantial effort in addressing the comments raised, but maintains the overall original assessment of the manuscript. Recent literature are full of examples of CO selective catalysts in the CO₂RR, including bulk metals, supported metal nanoparticles and single atom catalysts, so a Pd-containing catalyst (albeit with a low loading) hardly advances the field. On the mechanistic front, the authors' argument that this work provides novel mechanistic insights because they employed a number of characterization and computational methods does not hold water. With all the characterization and computational efforts, the proposed mechanism has been discussed repeatedly in the literature, i.e., introducing a second/third element changes the electronic structure and weakens the CO binding energy. The reviewer recognize the amount of effort dedicated to this work, however, is not optimistic about its impact on the relevant field. More detailed comments:

Response

We are grateful for the reviewer's continued engagement and the opportunity for us to further elucidate the merits of our work.

We acknowledge the wealth of literature on CO-selective catalysts in the CO₂RR, including those involving “*bulk metals, supported metal nanoparticles, and single atom catalysts*”. Nevertheless, we believe that our Cu₉₂Sb₅Pd₃ SAA catalyst has unique contributions to this field. Our research is not solely focused on surpassing the CO selectivity or stability benchmarks set by existing catalysts. Instead, our aim is to enhance the performance of more accessible and cost-effective Cu-based catalysts. **In this context, our Cu₉₂Sb₅Pd₃ SAA catalyst has demonstrated the highest CO selectivity and stability among Cu-based catalysts to date. Furthermore, compared to noble metal catalysts, our catalyst exhibits commendable selectivity and activity, as evidenced in **Fig. 2f** and **Fig. R1** of our manuscript.**

Fig. 2f | Comparison of state-of-the-art noble metal catalysts with the $\text{Cu}_{92}\text{Sb}_5\text{Pd}_3$ catalyst in the CO_2RR .

Fig. R1 | Comparison of the selectivities of state-of-the-art noble metal catalysts with the $\text{Cu}_{92}\text{Sb}_5\text{Pd}_3$ catalyst in the CO_2RR .

In addition, considering the reviewer's concern over the Pd component, which has been reiterated several times, we have previously compared the selectivity of our $\text{Cu}_{92}\text{Sb}_5\text{Pd}_3$ catalyst with that of Pd-based catalysts for CO_2 -to-CO conversion based on published literature (see **Fig. R2**). The results demonstrate that our $\text{Cu}_{92}\text{Sb}_5\text{Pd}_3$ catalyst exhibited greater CO selectivity than most of the Pd-based catalysts over a wide potential range, indicating its exceptional catalytic performance. Therefore, our $\text{Cu}_{92}\text{Sb}_5\text{Pd}_3$ catalyst is more competitive for the CO_2RR than most other Pd-based catalysts.

Fig. R2 | Comparison of the selectivities of $\text{Cu}_{92}\text{Sb}_5\text{Pd}_3$ and Pd-based catalysts in the CO_2RR . The catalyst references used were reproduced from: Pd NPs [J. Am. Chem. Soc. **137**, 4288-4291 (2015)], Pd-NC [Adv. Funct. Mater. **30**, 2000407 (2020)], Pd@Au [J. Am. Chem. Soc. **142**, 44, 18971-18980 (2020)], Pd_1Cu_1 [Sep. Purif. Technol. **320**, 124186 (2023)], 3.7 nm Pd [Nano Res. **10(6)**, 2181–2191 (2017)] and Pd Octahedra [Adv. Energy Mater. **9**, 1802840 (2019)].

On the mechanistic front, we respectfully disagree with the reviewer's assertion of a lack of novelty. Our study goes beyond the established understanding that introducing additional elements can modify the electronic structure and CO binding energy. We explored how Sb and Pd doping influences CO_2 adsorption, activation, and reduction pathways on the Cu surface, as well as the suppression of the HER and enhancement of stability. Additionally, our work proposes a general design principle for creating trimetallic SAAs with tunable properties for CO_2 reduction, offering novel insights that have not been previously reported.

In light of these considerations, we kindly asked the reviewer to re-evaluate our manuscript, recognizing the originality and significance of our contributions to the field of CO₂ electrocatalysis. We hope that the detailed explanations provided herein will facilitate a favorable reassessment of our work for publication.

Comment 1

Regarding the response to Comment 1, the authors stated “The co-introduction of two single-atom components collaboratively alters the electronic state of neighbouring copper atoms, making them more favourable for CO formation and less prone to the HER”, which is a reasonable hypothesis. The reviewer is curious that for many (or the majority of) Cu sites that are not close to Pd or Sb, how come they do not behave as typical metallic Cu sites, i.e., producing a wide range of C₂₊ products?

Response

We appreciate the reviewer’s curiosity regarding the behavior of Cu sites distant from Pd or Sb atoms.

As elucidated in our previous discussions, the Cu₉₂Sb₅Pd₃ matrix does not exhibit uniform activation across all Cu atoms due to the single-atom dopants Pd and Sb. Specifically, Cu sites that are remote from Pd or Sb atoms typically exhibit unaltered metallic Cu. Conversely, the catalytic behavior of Cu sites adjacent to Pd and Sb is significantly altered, exhibiting a much greater activity and lower barriers. This pronounced activity and lowered barriers at the modified sites effectively overshadows the behavior of the more distant Cu sites, making their contribution to the overall catalytic process.

For analytical purposes, we directly used pure Cu (211) to represent the unmodulated Cu sites in Cu₉₂Sb₅Pd₃ for comparative analysis. Based on our earlier theoretical calculations in **Fig. 4** and **Fig. S34** on Cu₉₂Sb₅Pd₃ and Cu (211), respectively, it is obvious that the kinetic barriers and theoretical activities are significantly inferior on these non-activated Cu sites. Due to the higher kinetic barriers and poorer activities of the non-activated Cu sites in the CO₂RR, the unaffected Cu atoms were prone to be less reactive, leading to their dormancy. As a result, the formation of C₂₊ products is not detected. In summary, our single-atom alloying strategy creates selective active Cu sites in the vicinity and consumes nearly all the CO₂ near the three-phase interface to produce CO, thereby suppressing the formation of C₂₊ products in the CO₂RR.

Fig. 4 | *c*, CO_2RR to CO on $\text{Cu}_{92}\text{Sb}_5\text{Pd}_3$. *d*, Theoretical rates of CO production on $\text{Cu}_{92}\text{Sb}_5\text{Pd}_3$ compared with three control samples.

Supplementary Fig. 34 | *a*, CO_2RR to CO on Cu (211).

In addition, we would like to mention that similar phenomena have also been reported for other Cu-based SAA catalysts with 3~5% single-atom dopants [*Nat. Nanotechnol.* **16**, 1386-1393 (2021); *Nat. Commun.* **12**, 1449 (2021)]. In these studies, the intrinsic properties of the entire Cu matrix were suppressed after single-atom doping, leading to the absence of multicarbon products in the CO_2RR . Consequently, it is plausible that our work detected no C_{2+} products following the incorporation of Pd and Sb into the Cu matrix, which is consistent with the reported phenomena.

Comment 2

Fig3 and Fig. S26 show inconsistent results of infrared and Raman results, i.e., CO band is absent on CuSbPd in Raman but present in infrared.

Response

To address the observed discrepancies between the infrared and Raman spectroscopy results, as shown in **Fig. 3** and **Fig. S26**, we offer a more cogent explanation grounded in the inherent differences between these techniques:

According to previous reports from Prof. Bingjun Xu and Prof. Karen Chan, ATR-SEIRAS and Raman spectroscopy are complementary methods that, due to their distinct mechanisms of enhancement and detection, may yield different observations when analyzing the same sample [*Nat. Commun.* **13**, 2656 (2022)]. It is well-documented that ATR-SEIRAS generally provides better signal-to-noise ratios and higher temporal resolutions within its spectral window than Raman [*Bone*, **139**, 115490 (2020)]. This is attributed to the larger IR cross-section, which is typically several orders of magnitude greater than that of Raman, allowing for more sensitive detection of species such as CO bands [*Journal of Analysis and Testing* **1**, 8, (2017)].

Furthermore, the surface enhancement effect of IR is less metal-specific than that of Raman spectroscopy. While Raman spectroscopy enhancement is largely limited to a few specific metals (such as Cu, Ag, and Au), ATR-SEIRAS can be conducted on a broader range of metallic surfaces, including Pt, Pd, and Ni. This broader range of applicability results in ATR-SEIRAS being more likely to detect signals from adsorbed species across various metal sites.

In our study, we utilized ATR-SEIRAS to enhance the detection of intermediates in the CO₂RR by depositing a polycrystalline Au film onto a Si crystal. This method significantly improves the signal intensity, explaining the prominent CO* signal observed in the ATR-SEIRAS experiments compared to that in the Raman spectra. To clarify, we have supplemented the details of the ATR-SEIRAS experiment in the **Supplementary Materials**.

The absence of the CO band in the Raman spectra for Cu₉₂Sb₅Pd₃ can be rationalized by considering that Raman spectroscopy tends to probe different subpopulations of adsorbates on weakly adsorbing surfaces, providing distinct information from that obtained on strongly binding surfaces. This suggests that the CO species detected by ATR-SEIRAS may not be the same as those detectable by Raman spectroscopy, leading to the observed inconsistency.

Given these technical distinctions, it is standard practice to employ both ATR-SEIRAS and Raman spectroscopy in electrocatalytic research to obtain a comprehensive understanding of the reaction intermediates and surface species involved in the CO₂RR. The combined use of these techniques allows for a more robust analysis, leveraging the strengths of each method to compensate for the other's limitations. Therefore, the differences observed between the two techniques in our study are consistent with their respective analytical capabilities and do not detract from the validity of the results obtained.

Reviewer 3

The resolution of all the issues I have raised will undoubtedly serve as a valuable source of inspiration for the researchers involved. Therefore, I strongly recommend accepting it for publication.

Response

We sincerely appreciate the reviewer's high evaluation of our study, as well as the important suggestions previously raised by the reviewer that greatly improved our work.

REVIEWERS' COMMENTS

Reviewer #1 (Remarks to the Author):

The authors have addressed my specific comments, albeit not entirely the general concern regarding the impact of the results reported. I will leave the editorial decision to the handling editors.